# The Application of Material Flow Cost Accounting in Waste Reduction

**Shaio Yan Huang [1], An An Chiu [2], Po Chi Chao [1,*] and Ni Wang [1]**

[1]   Department of Accounting and Information Technology, National Chung Cheng University, No.168, Sec. 1, University Rd., Minhsiung, Chiayi 62102, Taiwan; actsyh@ccu.edu.tw (S.Y.H.); s04527017@ccu.edu.tw (N.W.)

[2]   Department of International Business, Feng Chia University, No. 100, Wenhwa Rd., Seatwen, Taichung 40724, Taiwan; aachiu@fcuoa.fcu.edu.tw

\*   Correspondence: d05526001@ccu.edu.tw; Tel.: +886-9-3714-6164

**Abstract:** Due to the rise in environmental awareness, corporate companies have shifted their focus from an obsession with short-term profits to contemplating long-term strategies to achieve sustainable management. Effective use of resources is the primary indicator of this achievement. Fulfillment of corporate social responsibility and thinking beyond the regulatory aspects of corporate sustainable management are goals that have continually attracted attention worldwide. Material flow cost accounting based on ISO 14051, which was announced by the International Organization for Standardization (ISO), is a tool that can be used to achieve a balance between the environment and economy. We focused on using ISO 14051-based material flow cost accounting as an analytical evaluation tool from the perspectives of finance and accounting personnel. We conducted a case study on a flat-panel parts supplier to determine whether the efficient use of recycled glass could reduce company costs. The primary finding is that the film layer on recycled washed glass tends to be stripped during the production process, causing increased reprocessing costs and thus rendering the cost of renewable cleaning higher than that of reworking. This study revealed that the ISO 14051-based material flow cost accounting analysis constitutes a valuable management tool, thereby facilitating the promotion of sustainable development.

**Keywords:** sustainability; ISO14051; material flow cost accounting

## 1. Introduction

The rapid industry development, the depletion of natural resources, and the rise in environmental awareness have led corporate companies to shift their attentions from short-term profits to long-term strategies to achieve sustainable management and smooth progress into a new era. The World Commission on Environment and Development established sustainable development policies for future environmental and economic development, defining sustainability as a development model that meets the needs of contemporary people and protects the environment without comprising future competition. Sustainability indicators are a suitable means for assessing the development of production technologies and integration of business decisions [1]. Achieving an acceptable environmental performance has become a universal commitment for all organizations to maintain competitiveness. Environmental accounting (or green accounting) is an environmental analytical tool that measures and communicates the costs and benefits of the overall economic effects [2,3]. It is a process that involves collecting material-volume and cost information to identify the costs incurred by corporate companies in the pollution emission, waste treatment, and environmental protection. Currently, environmental accounting has garnered international attention and the disclosure of corporate environmental information is actively encouraged while theoretical frameworks and practical guidelines are gradually

developed. Environmental cost and ecological balance remain a topic of intense discussion in the field of environmental accounting. In 2007, the Environmental Protection Administration of Taiwan formulated a set of environmental-accounting guidelines, which are currently commonly adopted by Taiwanese corporations as the foundation for environmental-accounting practices.

Taiwan has a serious environmental pollution problem, such as air pollution caused by the power plants and pollution of water sources, which has a significant impact on the society, people, and enterprises. Even though many SMEs have environmental awareness, they do not have the knowledge required and/or the skills needed to apply the appropriate environmental tools [4,5]. One study indicated that the eco-efficient optimization of material flow cost management, which aims to reduce costs while simultaneously decreasing environmental influence, has become an explicit objective of practical and scientific activities and efforts. Although discussions have been frequent, little has actually been done to realize these goals [6]. Even in Germany and Japan, numerous companies remain ill-informed regarding the MFCA process and few have fully incorporated this practice into their operations [7]. However, Schaltegger et al. [8] used tools for sustainability management to examine large German companies and reported that more well-known environmental accounting tools are more likely to be implemented. Therefore, communication channels and publicity regarding the release of the ISO 14051: Material flow cost-accounting standard are crucial [7].

To standardize MFCA practices, a working group of the ISO Technical Committee ISO/ TC 207, Environmental Management, developed ISO 14051, which complements the ISO 14000 family of environmental management system standards, including life cycle assessment (ISO 14040, ISO 14044) and environmental performance evaluation (ISO 14031). The standard was published in the second half of 2011. Material flow cost accounting (MFCA) based on ISO 14051 is an environmental-management accounting (EMA) tool providing the analysis of environmental and economic balance [3]. ISO 14051 provides a general framework for material flow cost accounting. The ultimate goal of MFCA is to mitigate environmental problems and simultaneously improve the economic performance [9]. The concept of MFCA has been successfully applied in numerous industrial practices to improve waste-reduction decisions, such as in brewing companies in South Africa [3], small textile factories in Thailand [10], pharmaceutical companies in Japan [9], and small- and medium-sized enterprises in Malaysia [11]. These cases indicate that MFCA increases environmental sustainability and improves the overall economic performance of a company.

The purpose of this study is to use case companies to prove whether the introduction of MFCA management tools can be successfully promoted only by referring to the ISO14501 manual. In addition, by using the financial and cost data provided by the case company, this study explores whether the MFCA analysis tool is a set of stability tools, and whether it is possible to produce different results because of the different managers of operation analysis tools.

The case company is the first company in Taiwan fully implementing MFCA. The reason why the case company takes the lead to implement MFCA is to reduce the waste and pollution, which can assist the case company in reaching sustainability. Other companies in Taiwan only implement the MFCA in one factory or one product line, which makes it difficult to know the real effects of the implementation. Thus, understanding the implementation process and steps of the case company can provide suggestions to other companies.

The analysis of the case company's implementation of MFCA revealed that the film layer on recycled washed glass tends to be stripped during the production process, causing increased reprocessing costs that result in the cost of renewable cleaning being higher than that of using new materials. Because remanufacturing requires that used products serve as inputs, the success of remanufacturing depends on the efficiency with which used products are collected [12]. The present study does not focus on the process of collecting used products, which is a topic that can be addressed by manufacturers, retailers, or third parties in future studies. Despite uncertainty regarding whether the efficient use of recycled glass can reduce company costs and environmental loads, the case company can determine whether recycling is beneficial by evaluating the benefits of outsourcing recycled



materials following an analysis of ISO 14051-based MFCA. This study analyzes Taiwan CFI Company, the first company to receive ISO 14051 MFCA certification in Taiwan, which is recommended to get the business environmental protection award. The case company can be the role model of promoting business sustainability. The experience of applying MFCA can be the guidelines of the business in manufacturing and service industries, which are devoted to improving the sustainable environment and lowering the costs.

Studies on MFCA in Taiwan have primarily been based on environmental-management or industry cases. However, the basic information required for MFCA analysis is obtainable from the original cost-management accounting process. For the purpose of increasing the corporate resource efficiency and using natural resources more effectively and sustainably [13], analysis with MFCA is highly reasonable as the most prospective [14] and the only international standardized method [15]. Therefore, a cost-management accounting perspective was adopted, and a flat-panel parts supplier that had obtained ISO 14051-based MFCA certification served as a case in this study to describe implementation and teardown analysis methods, as well as investigate implementation results. Through this research, we hope to introduce a method as revolutionary as MFCA and help manufacturing companies to evolve in a changing environment and economy, overcome environmental damage inflicted by companies in a low-profit seeking era, and achieve mutual benefits for both the company and the environment. We expect that an analysis of ISO 14051-based MFCA will enable companies to consider all facets of MFCA and thereby become a valuable management tool that links companies to the environment and economy, as well as facilitates the promotion of sustainable development. The hypothesis of the research is whether the application of ISO 14051-based MFCA can assist the case company to analyze the cost, which helps manufacturing companies to evolve in a changing environment and economy.

The rest of this paper is organized as follows. In Section 2, we provide the review of the related literature. In Section 3, we give the background introduction and case analysis. Lastly, the final conclusions and recommendations are presented.

## 2. Literature Review

### 2.1. Material Flow Management

Adopting sustainable development while also retaining business competitiveness is a common goal in corporate communities. Because of this trend, a goal-oriented innovative management method is necessary for economic optimization and the reduction of material-related environmental pollution. Wagner and Enzler [6] explained that material flow management, which contributes to full utilization of the potential sustainable management of a company, provides a new framework for economic research and initiates standardization processes for sustainable management. Because material flow and its effects are direct causes of ecological problems, material flow management can be used to directly address the root of a problem and facilitate the reduction in the environmental pollution, which also leads to cost reduction [16].

Material flow management is focused on optimizing systems rather than a single product or material and is primarily characterized by interdisciplinarity and networking on the basis of a normative orientation [6]. The initial step of material flow management is goal setting. Because material flow management is a component of a superordinate management concept, the goals of material flow management arise from the normative and strategic goals of the company. In fact, the environmental-management tool with life-cycle-assessment orientation is designed to reveal the life-cycle–related effects of products and services on the natural environment. This approach helps to reduce environmental damage, but does not provide clear contributions to cost savings or corporate profits [16]. Traditional cost-accounting methods in the evaluation of material flow processes neglect the complexity of material and energy flow [16]. Environmental-management accounting has expanded

rapidly, and MFCA has become one of the most promising environmental-management accounting tools [17].

MFCA is an environmental-management accounting practice proposed by Professor Wagner from the Institut fur Management und Umwelt in Augsburg. MFCA aims to reduce material input through analysis. All input materials equal the amount of finished goods (positive products) plus that of generated waste (negative products). This equation represents the identification of material balance. When the number of positive products is constant, the number of negative products decreases, which reduces the environmental effect and amount of waste produced.

Environmental protection is critical for sustainability. Continuous investments in energy consumption and natural resource consumption, as well as manufacturing sectors and infrastructure, have had seriously harmful impacts on the environment [18]. Environmental accounting creates accountability for business entities in terms of their efforts to protect the environment in their corporate decisions [19]. Environmental accounting constitutes a tool for applying the sustainable development concept and now commands acceptance as a means of ensuring the preservation of the environment Environmental accounting information includes financial, environmental performance, and policy aspects, which are scattered in many parts of annual reports and social responsibility reports. The quality of disclosures can be characterized from aspects of comprehensiveness, reliability, and compliance (Masud, Bae, & Kim, 2017). In decision making, an organization considers different pressures from internal and external parties and attempts to legitimize the impact of its activities on the environment in the eyes of society and various pressure groups. Environmental accounting plays an active role in preparing, presenting, and analyzing environmental information for interested party holders, thus encouraging top management to improve environmental conditions [19].

*2.2. Material Flow Cost Accounting*

MFCA aims to reduce the material input through the analysis. All input materials equal the amount of finished goods (positive products) plus that of generated waste (negative products). This equation represents the identification of material balance. When the number of positive products is constant, the number of negative products decreases, which reduces the environmental effects and amount of waste produced. Nakajima [20] indicated that the production process in traditional cost accounting is a consumption process of economic value in which the added value of a product is valuated, and the consumption in an official production process is regarded as cost accounting. All monetary values of inputted resources are calculated in the production cost of a product as the output of the corresponding production process, meaning that a product should be burdened with all of the costs of utilized resources. Additionally, waste is not recognized in traditional cost accounting because it is considered to be unrelated to a value chain of various inputted resources (monetary value and physical volume), which are completely separated from production processes. By contrast, in MFCA analysis, wastes are not regarded as a process of value addition. Unlike traditional cost accounting, MFCA aims to recover the value of a product in a production process by equally valuing all of the outputs from a production process [20].

When using traditional cost-accounting methods, management authorities are mostly focused on the costs of manufacturing a product, which cover all of the costs incurred during the production process, including direct materials, labor, and production overheads. All of the costs incurred when manufacturing a product comprise the cost of the product. Irrespective of the amount of losses generated in an output process, the cost of losses is still allocated to the product cost.

In MFCA, quantity of input refers to the actual quantity of input remaining in the final product. Input in MFCA differs from input in a manufacturing and production process in that the quantity of input in the final product is the actual amount of input required to produce 100% yield without taking yield into consideration. Yield is generally considered in the production process, producing a standard value, which represents a standard dosage that is considered to be the target value. At the beginning of the production process, the quantity of allowable losses in management is considered

in addition to yield, and yield plus the quantity of allowable losses leads to the development of an ideal standard. The difference between a realistic standard and ideal standard is the loss examined in MFCA; accordingly, the hidden value in this loss can be identified.

In MFCA, waste is considered to be negative products. The main consequence of this assumption is that the manufacturing cost is used to produce not only the required products, but also the undesired negative products (waste). Thus, waste is considered to bear part of the processing cost of all upstream processing steps. Conventional cost-accounting practices typically overlook significant costs incurred by waste; therefore, the cost incurred by waste is known as the hidden cost. To reduce this cost and thereby improve the economic performance, waste recovery is an essential strategy for manufacturing companies [21].

Numerous studies worldwide have identified the advantages of MFCA. Several studies have identified the advantages of MFCA. Based on a case study of a Sri Lankan crepe rubber factory, Dunuwila et al. [22] adopted a cost-efficient and eco-friendly approach to improve the processing sector for natural rubber in the following three phases: 1. Quantifying factory resource use, economic loss, and greenhouse gas emissions through material flow analysis, MFCA, and life-cycle assessment; 2. developing proposals for viable improvement options; and 3. confirming the benefits of implementing the suggested improvement options. The results indicated that the proposed methods enabled 26% and 79% decreases in cost and global warming potential, respectively. Provided that the improvement options are viable, the results published by Dunuwila et al. [22] can be used to ultimately increase profits for the rubber company, reduce the amount of toxic gas released into the atmosphere, reduce water consumption, and enhance the company image.

Nitto is a leading Japanese diversified materials manufacturer that offers a wide variety of products, including tapes, vinyl, LCDs, insulation, and reverse osmosis membranes. With the support of the Japanese Ministry of Economy, Nitto was the first model company to use MFCA. Before introducing MFCA, Nitto had long been committed to enhancing the efficiency and mitigating the environmental effects of its operations. Nitto discovered that its environmental-accounting system was incomplete, primarily because only environmental protection costs were considered, and a few costs associated with environmental effects were still hidden. The environmental protection costs incurred through conventional environmental accounting only represented 1.6% of total sales in 2005. However, other costs, such as energy-consumption costs related to environmental effects, are usually regarded as production costs, which accounted for approximately 13.6% of total sales, substantially higher than that for environmental protection costs (1.6%). During the process of target-product selection, Nitto focused on a product that exhibited an upward market trend and had production lines that generated a considerable amount of material loss and consumed a substantial amount of energy during the manufacturing process. Through MFCA, Nitto could accurately assess its material losses and associated costs. This information became a driving force for Nitto to improve the material productivity of this production line. Hence, the implementation team established its improvement plans on the basis of a cost–benefit analysis using the physical and cost data under MFCA. The plans and countermeasures were subsequently planned and implemented. Nitto recognized MFCA as a method that was helpful for clarifying losses in terms of cost, enabled managers to set explicit improvement goals, and provided a means for identifying potential cost reductions and positive effects on the environment. Using a professional management method such as MFCA ensured increased competitiveness for achieving the goal of a sustainable operation (Manual on Material Flow Cost Accounting ISO14051, 2014).

Kokubu and Kitada [17] indicated that intensely competitive Japanese companies have identified further room for improvement by using MFCA because the loss concept in MFCA is different from the general concept of compliance in traditional business management. The value of MFCA-derived information is attributed to this concept of difference in loss.

Kokubu and Kitada [17] examined three cases of companies that had applied MFCA in the EMA trial of the Japanese METI and investigated how the case companies used MFCA to introduce

countermeasures to solve problems. The findings demonstrated that applying MFCA and DOE increased product quality and reduced the adverse environmental effects of the production process of the case-study company, thereby reducing the cost and strengthening competitiveness. Kokubu and Kitada [17] investigated how the case companies used MFCA to introduce countermeasures to solve problems. The first case study indicated that Tanabe Seiyaku introduced MFCA as a trial in 2001. Although most Japanese companies still use Microsoft Excel for MFCA calculations, Tanabe Seiyaku has already combined the enterprise resource planning system for the entire company with MFCA [23]. Through the integration of MFCA data, information on improvement activities is shared across the entire company. The second case is Canon Inc., one of the leading precise machine companies in Japan. The company introduced MFCA as a trial from 2001 to 2002, after which time Canon successively introduced MFCA at individual manufacturing plants on the basis of cooperation between the environmental department and plants. In a report on sustainable development, Canon also mentioned that through its workplace-centered environmental assurance system using MFCA, it is committed to reducing waste and cost. The last case study focused on Sekisui Chemical Co. Ltd., which introduced MFCA in 2004 and indicated that through MFCA activities, the company continues to lower and stabilize costs, verifying that from a business perspective, using an innovative MFCA management system is conducive to the development of manufacturing industries.

Chompu-inwai et al. [24] conducted a case study on a wood-product manufacturer in northern Thailand. An analysis of the company's production process revealed that almost 70% of the raw wood materials used were waste in the form of chippings, sawdust, off-cuts, and defects. The objective of the study was to discuss whether the application of MFCA can reduce material consumption and waste to the maximum extent. The results of their experiment were subsequently integrated into practical applications, reducing the amount of wood material loss incurred during the cutting process from approximately 69% to 54%. The findings demonstrated that applying MFCA increased product quality and reduced the adverse environmental effects of the production process of the case-study company, thereby reducing the cost and strengthening competitiveness.

In response to the rising cost of raw materials obtained from the natural environment, consistently high production costs, and the inevitable imposition of carbon or energy taxes by environmental agencies, the China Steel Corporation installed MFCA integrated with functions for managing resources, the environment, and waste. Findings obtained using this tool can serve as a basis for determining improvement options and provide clear insight into which types of cost and expenditure are the sources of maximum loss. Furthermore, these results reveal the flow and proportion of iron resources from the main products, recycling and reuse within plants, and waste gases to facilitate the formulation of improvement options.

One key difference is how you recognize the cost MFCA represents as a different way of management accounting. In conventional cost accounting, the data is used to determine whether the incurred costs are recovered from sales. It does not require determining whether material is transformed into products or disposed as waste. In conventional accounting, even if waste is recognized in terms of quantity, the costs to produce 'material losses' are included as part of the total output cost. On the other hand, MFCA, as explained before, focuses on identifying and differentiating between the costs associated with 'products' and 'material losses'.

In this way, 'material loss' is evaluated as an economic loss, which encourages the management to search for ways to reduce material losses and improve business efficiency. The figure below shows the difference between MFCA and Conventional Accounting. MFCA is the cost accounting system that can be installed in the ERP system. Therefore, there is no system integration issue of the MFCA system.

The aforementioned studies present cases of successful MFCA implementation. Thus, through the promotion of ISO 14051-based MFCA analysis, this study constructed a material flow model to determine the potential environmental and financial consequences of raw material and energy use and manufacturing processes. Other aims were to increase the transparency of material and energy use and manufacturing processes, as well as to highlight the loss of raw material input and waste-treatment cost

that have always been neglected. Creating a management mechanism can effectively strengthen green designs and enhance the research and development of clean processes, while also identifying methods for mitigating environmental effects and reducing costs, with the hope of achieving the optimal balance between the promotion of environmental conservation and economic growth. In traditional cost accounting, the cost of finished goods increases with the number of steps in the production process, and this cost increases incrementally. The physical volume of materials does not necessary accumulate with the number of steps in a production process because they may be cut and lost during manufacturing, thereby generating enough waste whose quantities plus the final output equal the total physical volume of materials inputted.

In traditional cost accounting, the basic calculation model involves allocating manufacturing costs to various products. Therefore, the value of wastes generated during the production process in most companies is "zero." Generally, little analysis is performed on waste, except for the waste management cost recognized in environmental accounting. The basic calculation model in MFCA involves using the same approach to allocate the manufacturing costs of finished goods (positive products) and waste (negative products) generated during the production process. Thus, the true value of generated waste (negative products) can be calculated, and a hotspot for improvement can be identified from this originally zero-value waste (negative products).

*2.3. ISO 14051 Based MFCA*

ISO 14051 defines MFCA as a "tool for quantifying the flows and stocks of materials in processes or production lines in both physical and monetary units" where 'materials' include energy and water [14]. These flows and stocks of materials are important because they pervade business practice [25]. The aim of MFCA is to provide information to management about opportunities for reducing material use and improving the monetary performance of businesses at the same time, representing an irresistible opportunity.

To facilitate implementation, ISO 14051 proposes several MFCA implementation steps, as delineated below. The level of detail and complexity of the analysis will depend on the size of the organization, nature of the organization's activities and products, number of processes, and quantity centers chosen for analysis, among other factors. These conditions make MFCA a flexible tool that can be applied in a wide range of organizations.

2.3.1. Implementation Step 1: Engaging Management and Determining Roles and Responsibilities (Clause 6.2, 6.3, ISO 14051:2011)

Successful projects usually start with support from the company's management; MFCA is not an exception. If the company management understands the benefits of MFCA and its usefulness in achieving an organization's environmental and financial targets, it is easier to gain commitment from the whole organization. In order to be effectively implemented, it is highly recommended that top management take the lead in MFCA implementation by assigning roles and responsibilities, including setting up an MFCA project implementation team, providing resources, monitoring progress, reviewing results, and deciding on improvement measures based on the MFCA results.

In general terms, management should be engaged in all phases of MFCA implementation and it is recommended that MFCA projects start with aggressive support from management, followed by a bottom-up approach on-site. In addition, successful implementation of MFCA requires collaboration between different departments within the organization. The reason why collaboration is needed is because different sources of information are required to complete MFCA analysis. Through engagement of the company management in the MFCA implementation process, the required expertise can be determined and the correct flow of information between all areas involved can be facilitated.

2.3.2. Implementation Step 2: Scope and Boundary of The Process and Establishing A Material Flow Model (Clauses 6.4, 6.5, ISO 14051:2011)

Based on collected material flow data, the MFCA boundary needs to be specified to clearly understand the scale of MFCA activity. During implementation, it is usually recommended that there is a focus on specific products or processes at the beginning, before then expanding implementation to other products. By implementing MFCA in steps, the analysis is simplified and better results can be achieved.

The boundary can be limited to a single process, multiple processes, an entire facility, or a supply chain. It is recommended that the process or processes that are chosen for initial implementation be the ones with potentially significant environmental and economic impacts. After specifying the boundary, the process should be classified in quantity centers using process information and procurement records. In MFCA, the quantity center is the part of the process in which inputs and outputs are quantified. In most cases, quantity centers represent parts of the process in which materials are transformed. If the material flow between two processes is the source of significant material loss, the flow can be classified as a separate material flow.

After determining the boundary and quantity centers, a time period for MFCA data collection needs to be specified. While MFCA does not indicate the period during which data must be collected for analysis, it should be sufficiently long to allow meaningful data to be collected and to minimize the impact of any significant process variation that can affect the reliability and usability of the data, such as seasonal fluctuations. Several historical MFCA projects indicate that the appropriate data collection period can be as short as a month, with a half-year or a year of data collection being the most common.

2.3.3. Implementation Step 3: Cost Allocation (Clauses 5.3, 6.8, ISO 14051:2011)

MFCA divides costs into the following categories:

- Material cost: cost for a substance that enters and/or leaves a quantity center;
- Energy cost: cost for electricity, fuel, steam, heat, and compressed air;
- System cost: Cost of labor, cost of depreciation and maintenance, and cost of transport;
- Waste management cost: cost of handling waste generated in a quantity center.

Material costs, energy costs, and system costs are assigned or allocated to either products or material losses at each quantity center based on the proportion of the material input that flows into the product and material loss. The material costs for each input and output flow are quantified by multiplying the physical amount of the material flow by the unit cost of the material over the time period chosen for the analysis. When quantifying the material costs for products and material losses, the material costs associated with any changes in material inventory within the quantity center should also be quantified. In contrast to material, energy, and system costs that are assigned to products and material losses proportionally, 100% of the waste management costs are attributed to material loss, since the costs represent the costs of managing this material loss.

2.3.4. Implementation Step 4: Interpreting and Communication MFCA Results (Clauses 6.9, 6.10, ISO 14051:2011)

MFCA implementation provides information such as material loss throughout the process, the use of materials that do not become products, overall costs, and energy and system costs associated with the material loss. This information brings about multiple impacts by increasing the awareness of the company's operations. Managers who are aware of the costs associated with material losses can then identify opportunities to increase efficiency in material use and improve business performance.

2.3.5. Implementation Step 5: Improving Production Practices and Reducing Material Loss through MFCA Results (Clause 6.11, ISO 14051:2011)

Once MFCA analysis has assisted an organization to understand the magnitude, consequences, and drivers of material use and loss, the organization may review the MFCA data and seek opportunities to improve the environmental and financial performance. The measures taken to achieve these improvements can include the substitution of materials; modification of processes, production lines, or products; and intensified research and development activities related to material and energy efficiency.

## 3. Research Method

This research adopted the case-study method using a flat-panel parts supplier as an example to analyze the implementation of ISO 14051-based MFCA by the case company and investigate the results. This method is most commonly used in exploratory research for which the relevant qualitative research techniques are convenient [26]. The case study method can disclose the problem and then systematically collect and analyze the problem in a scientific way to find the solution [27]. The importance of this case study is that the process of applying MFCA can provide suggestions for the future installation of MFCA. For example, the setting of the quantity center should be the same as the existing data record and periodic review framework to avoid ineffective use. Second, the historical data should be saved in the operating system, and the material flow currency should be quantitatively linked to the accounting, and the existing management system of each departments should be comprehensively integrated to facilitate the rapid introduction of the product. The analysis of material flow and energy use helps the company understand the relationship between material usage and cost, identify the hot spots of material and cost loss, improve the efficiency of data analysis, and systematically review the process to promote and strengthen the environmental cost management.

However, this is confidential financial and cost information within the company. Different companies have different cost data due to different processes and information system construction. In the absence of database or cross-company consistency data, we use case studies to conduct empirical analysis. Some management accounting-related research [28–30] uses case studies for the empirical analysis due to the lack of public information.

Although the case-study method provides a comprehensive understanding of a situation and causal relationships, it is still overly subjective and lacks objectivity. Thus, in our case analysis, we first described the implementation manuals adopted in Taiwan Industrial Development Bureau, Ministry of Economic Affairs, and then analyzed the case company by following the implementation methods specified by the manual to circumvent the effects of subjectivity on this research.

The financial information of the case company T is collected from the database of the Company T. The production process and cost structure are summarized from the database of company T. According to Clause 6.2, 6.3, ISO 14051:2011, successful projects usually start with the support of the company's management. MFCA is not an exception. Through engagement of the company management in the MFCA implementation process, the required expertise can be determined and the correct flow of information between all areas involved can be facilitated. Therefore, we also conducted in-depth interviews with the production manager, accounting manager, and other internal personnel to further understand the flow of the company's existing product line and cost structure. We interviewed five employees, including one production manager, one account manager, and three internal members of staff to increase the integrity of data collection through a discussion and observation of visits.

To facilitate implementation, ISO 14051 proposes five MFCA implementation steps, as delineated in the literature review section, which are listed in Section 2.3.

## 4. Case Analysis

### 4.1. Company Profile

The case discussed in the study is a Japanese company in Taiwan (hereafter referred to as "Company T"). Company T, established in 2001 with a capital of NT$15.3 billion, was a professional manufacturer and supplier of color filters and strategically collaborated with Company A in 2006. The plant consisted of two buildings, one of which specialized in the production of generation 4.5 and 5 color filters and the other of which produced generation 6 color filters. Company T was highly automated in that its equipment-production information was linked by the manufacturing execution system, which is managed by its industrial engineering department, to the ORACLE-ERP system (hereafter referred to as the "ERP system"). The company adopted the standard costing system as the cost-accounting method.

### 4.2. Decision for Implementing MFCA

As it strove to achieve sustainable development, Company T received the ISO14001 environmental-management system, ISO14064 greenhouse gas verification, carbon footprint and water footprint, and the ISO50001 energy management system certification before introducing the MFA system. To elucidate the environmental effects of its raw material and energy practices, Company T promoted the ISO14051-based MFCA in 2013, which could not only be used to comprehensively examine the transparency of the company regarding raw materials and energy use, but also to minimize its environmental effects while also reducing the costs incurred by losses. Ultimately, the company aimed to fulfill its social responsibilities in a low-interest-rate environment. Regarding research methods, we described the standard operating procedures for introducing MFCA analysis by using the "Introductory Manual for Industry Implementation of Material Flow Cost Accounting" from the Taiwan Ministry of Economic Affairs and the "Material Flow Cost Accounting Introduction.

### 4.3. The Initial Stage of Establishing a Project Group

While companies implement MFCA-based projects, companies often struggle with implementing the project and managing costs. To effectively implement MFCA, the collective participation of professionals among departments is necessary because the integration of the information for cross departments is crucial. For example, acquiring supervisory supports from the finance department and plant affairs department is greatly beneficial while implementing the MFCA. Establishing a project group is key in the initial stage of implementation. Project members include personnel from environmental safety, manufacturing, technical, plant affairs, and finance departments. The responsibilities of each department are as follows:

1. Finance department integrates the construction and cost information to construct a material flow model, compile reports, and document findings for future reference. It also provides valuable assistance in the implementation process, from preliminary planning to the compilation of an MFCA balance sheet. In the preparatory stage of MFCA implementation, a scope and time period must first be specified;
2. Environmental-safety department provides the information regarding the environmental effects of wastes, including types of the waste and waste-treatment activities;
3. Plant affairs department identifies types and methods of the energy application and quantifies the consumption of the energy, including the energy loss in each quantity center (QC);
4. Manufacturing and technical department analyzes the material balance in a material flow model and assists in providing the missing information.

*4.4. Scope Specification*

The key to successful MFCA implementation is to select the product or production line and time period in accordance with the following characteristics:

1.　　Product with the highest unit price;
2.　　Product or production line associated with the maximum material input;
3.　　Product or production line associated with the maximum energy consumption;
4.　　The simplest product variety;
5.　　Production line with a discernible scope;
6.　　Process that easily generates wastes.

*4.5. Quantity Center Selection*

A quantity center is a basic unit of MFCA calculation. Theoretically, MFCA is used to define the quantity center as work sections that generate loss during the process. The scope of the quantity center includes facilitating the collection and calculation of relevant data, but may lose its focus in practice when differentiating among the types and losses of negative product costs as a result of excessive variety. Conversely, a small scope for a quantity center may result in overly detailed and complex MFCA calculations, which impede and prolong the process of data collection and organization. Company T selected quantity centers according to their most detailed work sections.

*4.6. Cost Classification in MFCA*

MFCA is particularly effective for improving resource efficiency because it increases transparency through the visualization of material flow, relevant costs, and internal environmental effects [31]. Strobel and Redmann [32] indicated that four types of cost are classified under the concept of MFCA. The cost classification of Company T is described as follows:

1.　　Material Cost:

　　(1)　　Direct materials: Direct materials refer to materials inputted at the beginning of the process and remain in the final product. The direct materials of Company T include glass, BM resist, R resist, G resist, and B resist;

　　(2)　　Indirect and auxiliary materials: Indirect materials are added during an intermediary process. Auxiliary materials, such as detergents and solvents, are not directly required for manufacturing products, but are essential to the process. Auxiliary materials vary in type and account for less than 10% of the total cost. We therefore eliminated auxiliary materials in the current analysis.

2.　　Energy Cost (Water & Electricity cost):

The energy cost of Company T primarily consisted of water and electricity costs. Because the company had acquired Carbon Footprint, Water Footprint, and ISO50001 Energy management system certifications prior to introducing MFCA, it had implemented water-recycling measures and electricity-meter controls, which simplified the process of obtaining data on each quantity center. Water cost was based on the amount paid to the Taiwan Water Corporation, and electricity cost was based on the amount paid to Taiwan Power Company.

3.　　System Cost:

Other costs not related to materials, energy, and waste treatment were categorized as system costs. Depending on how Company T classifies its accounts, system costs include labor costs, repairs, depreciation expenses, rentals, tax, packaging fees, insurance premiums, and the cost of photomasks.

4. Waste-Management Cost:

Waste management cost refers to the costs incurred from treating the wastes resulting from manufacturing and production processes, including the costs and fees associated with air emissions, wastewater, and solid-waste treatment.

*4.7. Specification of a Time Period*

The specification of a time period is chiefly based on the period in which data can be easily collected and organized. This study analyzed one-year data to reduce the short-term abnormalities caused by short-term economic recessions. Because the implementation year is 2013, we selected data generated throughout 2012. Figure 1 depicts the time period of work orders specified by Company T.

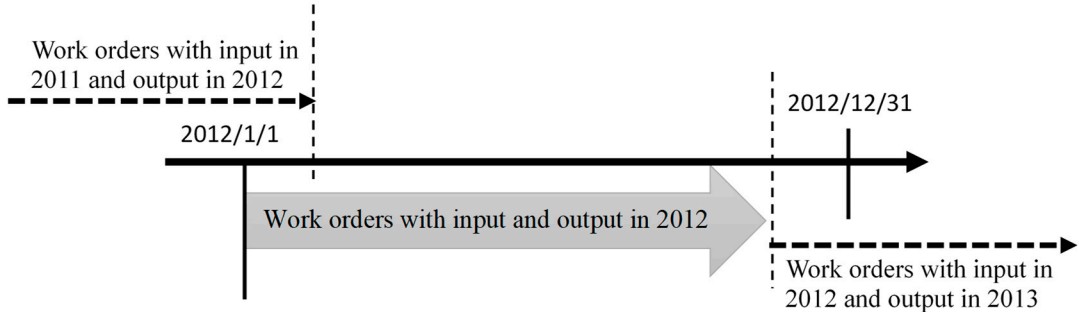

**Figure 1.** Time period of work orders specified by Company T.

Based on the aforementioned principles, the data of each quantity center were collected and organized as follows.

*4.8. Summary of Material Flow*

A material flow chart was compiled after the scope of a product or production line had been specified. Cost allocation in each quantity center was necessarily considered when drawing the material flow chart. The costs of each quantity center should not be more than or less than the allocation because over-allocation conceals loss items and under-allocation renders the process of data collection excessively time-consuming. Processes free of waste loss could be combined with other processes.

*4.9. Determination of Material Balance*

The materials of each quantity center are inputted according to the material flow chart and the output products are classified as either positive or negative products. Table 1 presents the process contents, input, output, and loss items of each quantity center. The input materials of each quantity center include direct materials, indirect materials, and all processed materials in a product. If auxiliary materials are excessively used and incurred wastewater-treatment costs, we advocated the incorporation of these materials in MFCA analysis. By integrating ERP and MFCA systems to obtain the cost of wastes, the management can acquire a material flow model chart (Figure 2) to improve the decisions [3].

**Table 1.** Description of each quantity center.

| QC | Process | Input | Output | Loss |
|---|---|---|---|---|
| QC1:<br>Unpacked Cleaner | Automated robot arm inputted material into the production line for production | Raw and recycled glass | Raw and recycled glass | Defective raw glass and recycled glass |
| | | Electricity | | Electricity |
| QC2:<br>BM processes | Coat a glass substrate with BM resist and project a photomask pattern onto the glass substrate, which is then exposed to photoresist agent and engraved to form a BM pattern. The primary function of a color filter was to enhance contrast and prevent light leakage and photoelectric flow | Raw and recycled glass | Partly finished BM goods | Defective raw glass and recycled glass |
| | | BM resist | | Defective BM resist |
| | | | | BM resist (switch outlet) |
| | | Auxiliary material | | Auxiliary material (including switch to cleaning and repair and maintenance) |
| | | Electricity | | Electricity |
| | | Water | | Water |
| QC3:<br>RGB processes | Expose R resist to an R-patterned photomask after it has been coated, and then use a developing agent to remove the unexposed part and thereby reveal the R pattern. Oven-dry the R resist to ensure that the pattern is drug resistant, and then repeat this process for the G and B resists. The colors red, green, and blue were separated by a BM. The primary function of the RGB resist was to transform black and white images into color images | Partly finished BM goods | Partly finished RGB goods | Defective raw glass and recycled glass |
| | | | | Defective BM resist |
| | | | | Defective RGB resist |
| | | R resist | | R resist (switch outlet) |
| | | G resist | | G resist (switch outlet) |
| | | B resist | | B resist (switch outlet) |
| | | Auxiliary material | | Auxiliary material (including switch to cleaning and repair and maintenance) |
| | | Electricity | | Electricity |
| | | Water | | Water |
| QC4:<br>POL and ITO processes | Grind the coated RGB glass substrate using a polyamide to flatten the photoresist layer. The purpose of the ITO process was to spread a layer of ITO film onto the flattened glass substrate to form a transparent conducting layer. | Partly finished RGB goods | Partly finished ITO goods | Defective raw glass and recycled glass |
| | | | | Defective BM resist |
| | | | | Defective RGB resist |
| | | ITO target | | Defect ITO target |
| | | Auxiliary material | | Auxiliary material |
| | | Electricity | | Electricity |
| | | Water | | Water |
| QC5:<br>PS processes | Subject the coated PS resist to light exposure, image-developing, and oven-drying to obtain substrate gap control for the required thickness to enhance contrast | Partly finished ITO goods | Partly finished PS goods | Defective raw glass and recycled glass |
| | | | | Defective BM resist |
| | | | | Defective RGB resist |
| | | | | Defective ITO target |
| | | | | Defective PS resist |
| | | PS resist | | PS resist (switch outlet) |
| | | Auxiliary material | | Auxiliary material (including switch to cleaning and repair and maintenance) |
| | | Electricity | | Electricity |
| | | Water | | Water |
| QC 6:<br>Inspection | Manually perform a visual inspection using an inspection machine | Partly finished PS goods | Finished goods | Defective raw glass and recycled glass |
| | | | | Defective BM resist |
| | | | | Defective RGB resist |
| | | | | Defective ITO target |
| | | | | Defective PS resist |
| | | Auxiliary material | | Auxiliary material |
| | | Electricity | | Electricity |

| QC | Process | Input | Treatment Method | |
|---|---|---|---|---|
| QC 7:<br>Waste treatment | Because waste at the output end are worthless (e.g., waste gas and wastewater discharge) or not within the scope of analysis (e.g., externally recycled materials), this QC describes input items and treatment methods | Scrapped glass | Outsource glass waste recycling treatment plants | |
| | | Partly finished goods that have been scrapped | Outsource ITO recycling treatment plants and outsource glass waste recycling treatment plants | |
| | | Output from a manufacturing process | Plant air-pollution control treatment, wastewater and recycled water treatment system, outsourced incineration treatment, outsourced recycling treatment, and external wastewater treatment (wastewater control) | |

QC: Quantity center.

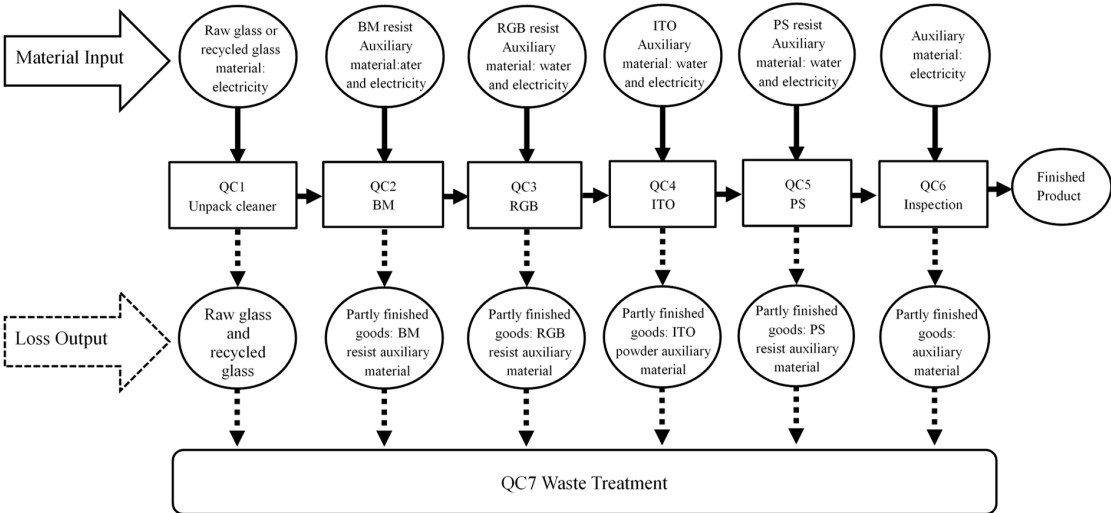

**Figure 2.** Material flow model for Company T.

*4.10. Quantification of Material Loss and Details (Material Balance Sheet for Each Quantity Center)*

The total material loss of all quantity centers was measured and summarized using material balance principles based on the material-flow model in Figure 2. Details and the monetary value of losses in all quantity centers were recorded. Subsequently, the total loss and subtotal loss details were compared to ensure data consistency. When a value cannot be measured, it is estimated using logic reasoning, and the unit of a material is expressed in kilograms or in terms of the area or volume if it cannot be converted into kilograms. All units of measurement should be the same to facilitate subsequent analysis and comparison.

*4.11. Calculation of Material Flow Cost*

The total quantity of a given material in each quantity center is calculated and multiplied by the unit purchase price to identify the hotspots with the highest loss rate.

1. Cost System Calculation Method Adopted by Company T

Company T was not only equipped with a highly automated production system and corresponding MES, but had also finished introducing and integrating an ERP system in 2010. The quantity centers are defined in the utmost detail and data collection primarily depended on support from the ERP system.

Cost calculation in the ERP system began with the bill of material. Two types of work orders were issued: a standard-order bill of material (realistic standard) and production-order bill of material (ideal standard). The difference between them was used to adjust uncertain losses during the production and to prepare for material distribution. The two types of work orders are described as follows:

(1) Standard-order bill of material: A standard-order bill of material is the accounting evidence and also consists of the amount of materials used in each quantity center, hours of labor, and other manufacture quantities of input materials required to produce a piece of color filter without any substantial loss ring costs;

(2) Production-order bill of material: It is almost impossible that no loss is incurred during the production process. Therefore, production-management staff adopt a standard-order bill of material as the basis for adjusting input quantity according to the status of a production line. Adjustments are most commonly made to yields. On a production-order bill of material, the quantity of usage is also the basis for calculating material stock and the quantity of materials to be purchased. Cost accounting is based on the data provided on a production-order bill of material, which documents the production start time, actual usage in each quantity center, and

the amount of loss. In the case of actual usage, cost is calculated by multiplying the bill of material-based input quantities in each quantity center by the standard unit price. Standard unit price is the unit price of the most recently approved order. The actual cost of each quantity center is summarized in Table 2. In the following section, adjustments to figures in the table are simulated. After a work order has been completed and inventoried, a work order record sheet is completed, as displayed in Table 3.

**Table 2.** The actual cost for each quantity center.

| Item | | | | QC 1 | | QC 2 | | QC 3 | | QC 4 | | Total | |
|------|------|------|----------------------|---------------------|--------------------|---------------------|--------------------|---------------------|--------------------|---------------------|--------------------|---------------------|--------------------|
| Type | Name | Unit | Standard Unit Price | Quantity of Usage | Monetary Amount | Quantity of Usage | Monetary Amount | Quantity of Usage | Monetary Amount | Quantity of Usage | Monetary Amount | Quantity of Usage | Monetary Amount |
| Material | Glass | Piece | 2800 | 303 | 848,400 | | - | | - | | | 303 | 848,400 |
| | Resist 1 | Kg | 3200 | | - | 0.8 | 2560 | | - | | | 0.8 | 2560 |
| | Resist 2 | Kg | 1800 | 0.7 | 1260 | | - | | - | | | 0.7 | 1260 |
| | Resist 3 | Kg | 1500 | | - | | - | | - | 0.8 | 1200 | 0.8 | 1200 |
| | ITO | Kg | 21,000 | | - | | - | 0.2 | 4200 | | - | 0.2 | 4200 |
| Labor | Direct labor | Hour | 125 | 3.5 | 438 | 4 | 500 | 2 | 250 | 3.5 | 438 | 13 | 1625 |
| | Indirect labor | Hour | 165 | 3 | 495 | 3 | 495 | 3 | 495 | 3 | 495 | 12 | 1980 |
| Manufacturing Costs | Depreciation expense | Area/Square | 10,000 | 10 | 100,000 | 30 | 300,000 | 20 | 200,000 | 20 | 200,000 | 80 | 800,000 |
| | Consumables | Hour | 800 | 3.5 | 2800 | 4 | 3200 | 2 | 1600 | 3.5 | 2800 | 13 | 10,400 |
| | Other costs | Hour | 600 | 3.5 | 2100 | 4 | 2400 | 2 | 1200 | 3.5 | 2100 | 13 | 7800 |
| | Electricity | KWH | 4500 | 3.5 | 15,750 | 4 | 18,000 | 2 | 9000 | 3.5 | 15,750 | 13 | 58,500 |
| Total | | | | | 971,243 | | 327,155 | | 216,745 | | 222,783 | | 1,737,925 |

Note: The examples for QC2 are shown in Table A1 of Appendix A, respectively.

**Table 3.** Quantity of materials used in a work order.

| [Work Order: A001] | | | MFCA Terms | Realistic Standard | | Yield | Ideal Standard | Consumption Quantity | | Order Difference | Loss |
|---|---|---|---|---|---|---|---|---|---|---|---|
| Type | Name | Unit | Standard Unit Price | Standard Usage | Standard Monetary Amount | Yield | Usage in Order | Actual Quantity of Usage | Monetary Amount Equivalent of Actual Amount of Usage | Monetary Difference | NG Quantity |
| Material | Glass | Piece | 2800 | 300 | 840,000 | 99% | 303 | 303 | 848,400 | 8400 | 3.0 |
| | Resist 1 | Kg | 3200 | 0.3 | 960 | 98% | 0.306 | 0.8 | 2560 | 1600 | 0.5 |
| | Resist 2 | Kg | 1800 | 0.6 | 1080 | 99% | 0.606 | 0.7 | 1260 | 180 | 0.1 |
| | Resist 3 | Kg | 1500 | 0.6 | 900 | 97% | 0.618 | 0.8 | 1200 | 300 | 0.2 |
| | ITO | Kg | 21,000 | 0.1 | 2100 | 85% | 0.115 | 0.2 | 4200 | 2100 | 0.1 |
| Labor | Direct labor | Hour | 125 | 10 | 1250 | 80% | 12 | 13 | 1625 | 375 | 3.0 |
| | Indirect labor | Hour | 165 | 11 | 1815 | 80% | 13.2 | 12 | 1980 | 165 | 1.0 |
| Manufacturing Costs | Depreciation expense | Area/Square | 10,000 | 80 | 800,000 | 100% | 80 | 80 | 80,000 | - | 0.0 |
| | Consumables | Hour | 800 | 11 | 8800 | 100% | 11 | 13 | 10,400 | 1600 | 2.0 |
| | Other costs | Hour | 600 | 12.5 | 7500 | 100% | 12.5 | 13 | 7800 | 300 | 0.5 |
| | Electricity | KWH | 4500 | 10 | 45,000 | 100% | 10 | 13 | 58,500 | 13,500 | 3.0 |
| Total | | | | | 1,709,405 | | | | 1,737,925 | 28,520 | |

In the process of MFCA calculation, a realistic standard refers to the cost, as stated in a standard bill of material, required to produce a piece of color filter without incurring loss. An ideal standard refers to the anticipated input quantity under consideration for yield. Quantity of consumption refers to the actual quantity of the input. The difference between the realistic standard and quantity of consumption is the difference in work order. Company T performed an analysis by multiplying the realistic standard quantity of usage by the annual average unit price. The cost required to produce a piece of color filter is regarded as the basis for calculating positive products, and the remaining costs are regarded as losses. This approach differs considerably from the calculation methods generally used in cost accounting.

In the initial stage of implementation, Company T attempted to use the cost-accounting method based on the material inventory sheet of a standard cost system and then allocated order differences to each material cost. However, MFCA analysis first appealed to the rate of losses and subsequently based cost allocation on the monetary amount or quantity of each production, which does not significantly influence the percentages of positive and negative products.

2. Material-Flow Cost-Accounting Analysis

The aim of the MFCA technique is to assess material and energy flow in material flow and monetary evaluations during the production process [9]. MFCA is a tool that captures material flow and monetary evaluations. The most substantial difference between cost accounting and MFCA is in the calculation of loss. Cost accounting of company T incorporates the cost of losses into the cost of a finished product. For example, the quantity of consumption in Table 3 was recorded in terms of the quantity of losses on a work order form, but the cost of loss was "zero." By contrast, MFCA involves the separate calculation of the sum of the cost of losses and determination of the causes for these losses. The focus of MFCA is on recognizing waste as nonmarketable products. In this study, we consolidated work orders for an entire year and listed all of the inputted materials, ascertaining "Input = positive products + negative products" to obtain a material flow model record sheet for Company T.

3. Material Input and Loss Calculation

Material quantity is calculated using the total quantity of materials in a work order that has both input and output in the same year within the scope of the specified time period. Calculations were based on the scope of the calculation in Table 4.

**Table 4.** Scope of calculation.

| QC | Quantity of Material Input (INPUT) | Usage of Positive Products in Each Resist | Quantity of Resist Lost | Quantity of Losses | Input Cost and Cost of Losses |
|---|---|---|---|---|---|
| QC1: Unpacked Cleaner | Quantity of glass inputted in QC1 is the total quantity of glass inputted in 2012 (because QC1 only has glass input and no other material input, and the input of a piece of glass produces a piece of glass filter, which is characteristic of the company's product) | | | Quantity of glass lost = quantity of defective glass at a station | Quantity × unit price = input and loss costs |
| QC2: BM Processes | Quantity of positive products in QC1 OUTPUT + Total quantity of new materials input in QC2 | Sum of the quantity of glass positive products × the realistic-standard quantity of resist usage in each work order | Total number of resist input − the value for the usage of each resist positive product | Quantity of glass lost = quantity of defective glass at a station | Quantity × unit price = input and loss costs |
| QC3: RGB Processes | Quantity of positive products in QC2 OUTPUT + Total quantity of new materials input in QC3 | Sum of the quantity of glass positive products × the realistic-standard quantity of resist usage in each work order | Total number of resist input − the value for the usage of each resist positive product | Quantity of glass lost = quantity of defective glass at a station | Quantity × unit price = input and loss costs |
| QC4: POL Processes and ITO Processes | Quantity and monetary amount of each material used in the QC thereafter | | | | |
| QC5: PS Processes | Quantity and monetary amount of each material used in the QC thereafter | | | | |
| QC 6: Inspection | Quantity and monetary amount of each material used in the QC thereafter | | | | |
| QC 7: Waste Treatment | Quantity and monetary amount of each material used in the QC thereafter | | | | |

The total quantity of each material in each quantity center is calculated and multiplied by the unit purchase price to produce the MFCA balance sheet and identify hotspots with the highest loss rate.

*4.12. Summary of Energy and System Costs*

Companies establish the cost center to which manufacturing costs can be allocated. Generally, the collection and organization of system and energy costs are based on data regarding manufacturing costs. Therefore, the cost center of a company can use manufacturing costs as a source of data regarding energy and system costs. Quantity centers in MFCA encompass system and energy costs, which are allocated to quantity centers according to the quantity of positive and negative products. Companies have begun to install electronic meters in their production lines to control electricity consumption, thereby conserving energy and reducing carbon emissions. If energy costs can be determined for individual quantity centers, then the results can be effectively analyzed.

1. Energy Cost (Water & Electricity Cost) Calculation

Company T selected the sixth generation plant, primarily because the production line is equipped with unique water and electricity meters, which facilitated the collection of energy data. However, not all stations had an electricity meter. Therefore, we still used input quantity as the basis of the allocation. Table 5 presents details of the calculation method. As shown in Table 5, Input Cost of Each QC that Cost allocation and calculation for QCs that have inputted water resources, as displayed in Table 6 = Total quantity of glass input at a station (including the number of times stripped glass was reworked) × the unit cost of tap water.

**Table 5.** Energy cost (Water & Electricity Cost) calculation method.

| Energy | Usage | Unit Cost | Input Cost of Each QC | Cost of Losses in Each QC |
|---|---|---|---|---|
| Tap water | Data regarding tap-water consumption were obtained from the quantity and monetary amount indicated by the Taiwan Water Corporation, but these data excluded the consumption of pure water | Unit cost of tap water = total monetary amount for 2012/total water consumption | Cost allocation and calculation for QCs that have inputted water resources = Total quantity of glass input at a station (including the number of times stripped glass was reworked) × the unit cost of tap water | = Total quantity of glass input at a station (including the number of times stripped glass was reworked) × the unit cost of tap water |
| Electricity | The plant owned by Company T supplied electricity to its two plant buildings; therefore, electricity consumed by the plant was allocated according to the ratio of electricity consumption by the two plant buildings. Subsequently, total electricity consumption was allocated to each QC, and costs were calculated as follows: Total electricity consumption = (amount of electricity consumption by a production line + amount of electricity allocated to plant buildings) in 2012 | Unit cost of electricity = (monetary amount for a production line indicated by the Taiwan Power Company + the monetary amount allocated to plant buildings) in 2012/total electricity consumption | Cost allocation and calculation of QCs that have inputted electricity = total quantity of glass input at a station (including the number of times stripped glass was reworked) × the unit cost of electricity | = Total quantity of defective glass at a station (including the number of times stripped glass was reworked) × the unit cost of electricity × the number of stations passed |

**Table 6.** Energy-cost calculation of Company T. (Unit: NT$).

| Item | Work Section | Input Quantity | Monetary Input | Average Unit Price | Cumulative Input Quantity | Cumulative Monetary Input |
|---|---|---|---|---|---|---|
| Tap Water | QC1 INC | | | | | |
| | QC2 BM | 53,260 | 698,425 | 13.11 | 53,260 | 698,425 |
| | QC3 RGB | 222,398 | 2,916,395 | 13.11 | 275,658 | 3,614,820 |
| | QC4 POL/ITO | 301,265 | 3,950,633 | 13.11 | 576,923 | 7,565,453 |
| | QC5 PS | 350,053 | 4,590,404 | 13.11 | 926,976 | 12,155,857 |
| | QC6 INS | 344,913 | 4,523,001 | 13.11 | 1,271,889 | 16,678,858 |
| Tap Water Subtotal | | 1,271,889 | 16,678,858 | 13.11 | | |
| Electricity | QC1 INC | 2,905,791 | 4,217,049 | 1.45 | 2,905,791 | 4,217,049 |
| | QC2 BM | 20,315,148 | 29,482,497 | 1.45 | 23,220,939 | 33,699,546 |
| | QC3 RGB | 55,192,669 | 80,098,737 | 1.45 | 78,413,608 | 113,798,283 |
| | QC4 POL/ITO | 93,347,887 | 135,471,756 | 1.45 | 171,761,495 | 249,270,039 |
| | QC5 PS | 108,138,933 | 156,937,362 | 1.45 | 279,900,428 | 406,207,401 |
| | QC6 INS | 109,630,206 | 159,101,582 | 1.45 | 389,530,634 | 565,308,983 |
| Electricity Subtotal | | 389,530,634 | 565,308,983 | 1.45 | | |
| Total | | 390,802,523 | 581,987,841 | 1.49 | | |

Note: Figures adjusted through simulation. The examples for Water cost are shown in Table A2 of Appendix A, respectively.

2. System Cost Calculation

System costs includes labor, depreciation expenses, consumables, and other expenses. With the exception of labor costs, system costs are fixed costs. Table 7 presents the system cost calculation method.

**Table 7.** System cost calculation method.

| Cost | Calculation Method | Unit Cost | Input Cost of Each QC | Cost of Losses in Each QC |
|---|---|---|---|---|
| System Cost | System cost = total production costs for a work order that has both input and output in 2012 − (Material cost + energy cost + waste treatment cost) | Unit cost of system cost = system cost/(the quantity of glass input at a station + the number of times stripped glass was reworked at each station) | Cost allocation and calculation for each QC = quantity of glass input at a station (including the number of times stripped glass was reworked) × the unit cost of the system | = Quantity of defective glass at a station (including the number of times stripped glass was reworked) × the unit cost of the system × the number of stations passed |

*4.13. Waste-Treatment Cost Calculation*

The income from scraps sold was deducted from the cost expended on handling waste to obtain the sum of the entire waste treatment cost. Because each quantity center generated waste, the waste could be viewed as a quantity center for calculation to facilitate data collection. Waste treatment cost includes waste-treatment fees, air-pollution controls and handling fees, and wastewater control and handling fees, all of which also cover outsourcing fees and the deduction of scrap income. Waste treatment cost reflects 100% loss and is therefore not allocated to product costs.

*4.14. Development of Improvement Plans*

After establishing an MFCA calculation model, material balance sheet, and MFCA balance sheet, we identified negative product costs and types, reasons, and quantity of losses; proposed improvement topics; determined improvement targets; investigated the feasibility of improvement methods; and implemented the improvement methods on the basis of predicted improvement effectiveness. MFCA is essentially a practice that improves a manufacturing site. MFCA is a form of analysis and planning, and daily activities conducted at a manufacturing site are practices. Thus, blending MFCA and daily activities ensures that the results of MFCA analysis are goal-oriented and can therefore provide concrete benefits rather than vague and superficial results. When numerous concerns must be addressed, MFCA serves as an effective prioritization tool.

1. Results of Material Flow Cost-Accounting Analysis

We performed an MFCA analysis on each quantity center based on the information provided in the aforementioned material flow model record sheet and subsequently produced an MFCA balance sheet.

2. Findings Based on an Analysis of the Material Flow Cost Accounting Implementation of Company T

Because the original product yield of Company T was 99.1%, the cost of the negative products was low. Of all costs, the material cost (67.1%) was the primary input of a product, in which the cost of glass materials accounted for 51% of the material cost. Through analysis, we discovered that lowering the glass-stripping rate resulted in reductions of other material, energy, and system costs required for reworking. Additionally, recycled glass is associated with a high stripping rate. Whether the sum of this cost and the cost of renewable cleaning is lower than the cost of reworking should be re-evaluated to determine the benefits of outsourcing recycled materials. Even though MFCA is considered an effective tool for reducing waste and environmental effects, its procedures are confined to recycling processes and recycled materials involving a production system. Therefore, product designs should aim to lower the cost of byproducts and fully utilize the materials of a production department to ultimately reduce the amount of waste produced [10].

Regarding the energy cost, problems related to climate change are integral to the sustainable development of Company T. The analysis indicated a loss rate of 3.09% in the energy cost. In response, the company can first improve its power supply by installing smart electricity meters and developing energy-resource management systems to track energy usage and then specify goals to promote environmental sustainability and emission reduction. In 2012, Company T introduced the goal of reducing the emissions in 2010 by an additional 25% by 2015.

Regarding the waste treatment cost, Company T can improve its efficiency in the process of collecting volatile organic compounds and reduce its hazardous air pollutant fugitive emissions by increasing the process efficiency of its terminal air-pollution control equipment. In 2012, the company improved its method for wastewater treatment by investing in an anaerobic wastewater-treatment system and increasing wastewater-treatment efficiency.

The analysis revealed that the internal process management of the company could still be improved. For example, the collected data indicated that the costs and ratios of losses were low, and the materials lost had no value and could therefore be treated as waste. Consequently, we were unable to determine whether the quantity of recognized waste was the same as the ideal quantity of waste. Because the losses could not be accurately determined, the realistic standard usage implied in the bill of material might include the reasonable loss standard. Therefore, this process must be further improved. Additionally, because the records of reworking are not properly managed, the question of whether reworking should be incorporated into process management can be considered in the next stage of improvement.

*4.15. Improvement Implementation*

In the initial stage of implementing improvements, the following steps can be taken: become familiar with the MFCA operating process, remain up-to-date regarding the logic and methods of data collection, and then initiate improvement plans for MFCA. In performing a detailed analysis, the management indicators of a manufacturing site (e.g., finished product rate, defect rate, and availability or uptime) can be combined to help define a standard value or improvement target and achieve these standards or targets. MFCA focuses on the analysis of informative data, and specifically on the application of data following analysis. The operation of MFCA essentially involves the application of a plan–do–check–act cycle, which is the key to project improvement and continual improvement for many corporations. A clearly defined scope of responsibility and the participation of executives and interdepartmental professionals are the pillars for implementing short-term, mid-term, and long-term MFCA improvement plans. MFCA can provide companies with informational feedback, including feedback on financial performance, for operating the planning and completion phases of the plan–do–check–act management cycle, considering subsequent improvement targets and decisions, and defining new corporate improvement indicators. Material flow is an integral component of management for two reasons: from an environmental perspective, material flow reflects the direct influence of an organization on the environment; and from an economic perspective, it helps to reduce the production cost, thereby enhancing the corporate financial performance [25]. Thus, this phenomenon proves that MFCA is a tool that enhances the corporate resource efficiency and thereby facilitates the actualization of sustainable development [31].

## 5. Conclusion and Recommendations

Companies have begun to increasingly focus on how they can align requirements for social sustainability and competitiveness [33]. When companies cease to pursue only individual sustainable operations, they inevitably regard the environment as a component of corporate development [34]. MFCA has expanded the scope of conventional accounting to include the environment and the economic effects of corporate sustainability and eco-efficiency concepts [21,35], thereby rendering goal management increasingly more scientific and systematic. This study assists companies to examine the process of recycled glass production by using ISO 14051-based MFCA and investigate whether a company can reduce costs by using recycled glass and mitigate the pollution and improve the companies' environmental performances.

There is no previous paper discussing the MFCA implementation steps and providing suggestions for reducing the cost in Taiwan. One of the reasons for this is that the number of certified companies in Taiwan remains low, and whether these companies continue to conduct analyses after receiving certification is unknown. The reason that the case company successfully implements MFCA is the attitude of its chief executive officer. In the second year of implementation, the case company performed an analysis of the entire plant and discovered that its wastewater-treatment expenses were positively correlated with resist loss, chemical usage, and water consumption. In the one month that did not exhibit a positive correlation, the company discovered that the sluice gate controlling wastewater discharge was damaged, which was causing a considerable increase in wastewater-treatment expenses because of the direct discharge of untreated wastewater. Although MFCA does not yield immediate effects regarding an increase in company profits, this practice demonstrates another type of long-term corporate value that contributes to effective cost reduction in low-profit environments. Compared with previous research, this paper not only introduces the advantages and effects of implementing the MFCA, but also lists the key action plan when applying the MFCA. This can help future companies to have a complete idea in assessing whether to implement the MFCA.

To bolster competitiveness, companies have continually invested monetary resources in the adoption of environmentally friendly practices. However, questions are often raised during this process regarding whether such an investment is truly beneficial for the environment or corporate development, and some companies remain focused on acquiring certifications to boost sales performance. Currently,

introducing MFCA helps companies to obtain not only certificates that contribute to sales, but also rewards in the form of cost reduction due to environmental initiatives.

Team composition, interpersonal communications, and the efforts of revolutionists are factors that contribute to the success of MFCA implementation. However, a potential barrier for improving MFCA is the inability of senior managers to solve performance-management problems [11]. Government authorities in Taiwan have begun to follow the steps implemented in Japan in terms of encouraging companies to introduce ISO14051-based MFCA analysis. Personnel in contact with companies primarily include those from environmental-safety, quality-control, or plant affairs departments. However, the basic information required for the analysis is generally managed by the finance and accounting departments. Therefore, the implementation of MFCA should collect, analyze, compile, and describe data from the perspectives of financial and accounting specialists to alert these specialists to the difference between conventional cost accounting and MFCA. After a balance sheet for basic analysis has been obtained, it should then be administered to plant personnel for hotspot improvements.

We determined that the original rate of product yield for the case company was 99.1%, indicating that the costs of negative products were low. Of all costs, the cost of glass materials accounted for 51% of the material cost (67.1%). During analysis, we discovered that reducing the glass-stripping rate resulted in a reduction of other material, energy, and system costs required for reworking. However, recycled glass is associated with a high stripping rate. Whether the sum of this cost and the cost of renewable cleaning is lower than the cost of reworking should be re-evaluated to determine the benefits of outsourcing rework. The analysis also indicated a loss rate of 3.09% in the energy cost. In response, the company can improve its power supply by installing smart electricity meters and developing energy-resource management systems to monitor energy usage. Therefore, this study verified that MFCA analysis both provides insights regarding areas that require improvement in the internal process management of the company and helps the company to achieve sustainable development with the goal of promoting environmental sustainability and reducing emissions. The primary finding is that the film layer on recycled washed glass tends to be stripped during the production process, causing increased reprocessing costs and thus rendering the cost of renewable cleaning higher than that of reworking.

By observing the case company's promotion of the ISO14051 MFCA tool, we can confirm that the inter-departmental flow of information is more accurate according to the steps in Chapter 2.3.1, which helps organizations better understand the application of materials and energy. Because the scope of sharing was expanded, it helped to improve the company to reduce the implementation of operating procedures. Through this study, we also found that managers could understand the MFCA tool more easily, and also helped the organization to better understand the application of materials and energy. That is to say, even if different managers operate MFCA tools, the results of this study are not very different from those calculated by the case company. Therefore, it is an analytical management tool that can be popularized.

The MFCA analysis tool is effectively optimizing the cost management system in reducing the cost, especially when the limited resources are input by the business. Thus, the implementation of the MFCA analysis tool receives more benefits than cost spending. The academic contributions of the study are as follows. The study promotes the application of MFCA to SMEs because the case company sets an example to other SMEs that MFCA can be embarked on with relative ease, and indicates perhaps an inexpensive cost of implementation. The research fills the research gap by providing the steps and suggestions of the successful application of an ISO 14051-based MFCA case in Taiwan. The case company utilizes MFCA to analyze the recycling process of a film layer, which can actually reflect the cost of the waste. The results can be applied to review the recycling process of the case company to effectively reduce the costs. The analysis can achieve the goal of reducing the costs and increasing the efficiency and set the model for promoting the sustainability.

The practical contributions of the study are as follows. This study analyzes Taiwan CFI Company, the first company to receive ISO 14051 MFCA certification in Taiwan, which is recommended to get the business environmental protection award. The case company can be a role model for promoting

business sustainability. The experience of applying MFCA can act as guidelines for businesses in manufacturing and service industries, which are devoted to improving the sustainable environment and lowering the costs.

The MFCA analysis shows that the recycling process increases the reprocessing cost, which is not beneficial for the case company to promote the sustainability. It provides, based on the case company, improvement directions and guidelines for waste recycling after the application of MFCA. The results can utilize the Material Circularity Indicator, developed by the Ellen MacArthur Foundation, to analyze the actual recovery rate of the materials and the life or frequency of the product use, which can be the reference for the improved design of the second-hand recycling process system.

Recommendations

MFCA is still in the nascent stage of implementation in Taiwan. Companies remain ill-informed with respect to the process of MFCA analysis. Furthermore, support services in Taiwan are primarily grounded in science and technology and lack professional knowledge on finance and accounting. This results in a context that is characterized by different cost concepts, missing data, and a lack of multidisciplinary talents, which greatly impedes MFCA implementation. However, because finance and accounting units primarily hold the controlling power and a strong subjective awareness, when implementation is governed by these units, the result is often a phenomenon that deviates from the site of implementation. Therefore, departmental support is necessary during project development. In the early stages of MFCA implementation, finance and accounting personnel should not be overly focused on balancing data and account analyses because analysis results are not affected by a one-dollar difference. Moreover, the purpose of MFCA is not solely to reduce the consumption of a single company. MFCA is ultimately aimed at achieving green activities on a global scale through the integration of upstream, midstream, and downstream processes.

The limitations of this case study should be considered. Because remanufacturing requires used products to serve as the input, the success of remanufacturing largely depends on the efficiency with which used products are collected [12]. Recycled glass is a used item; therefore, the reliability of recycled and washed glass in the recycling system of a production process should be extensively evaluated before a study is conducted. The present study was not focused on the process of collecting used products. The reliability of recycled glass in the recovery process should be evaluated before the study. This study does not focus on the collection process of used products. The evaluation of the effectiveness and reliability of the recycling process requires further steps, which can be carried out by manufacturers, retailers, or third parties. That is much more suitable for the follow-up research. Further steps are required to assess the effectiveness and reliability of a recycling environment, and in future research, these could be implemented by manufacturers, retailers, or third parties, which is more suitable for subsequent studies.

The case company of this study was highly automated in that its equipment-production information was linked by an MES that was managed by its industrial engineering department and integrated into the ORACLE-ERP system. With the advent of Internet of Things, various technologies have led to the development of automated systems and digital information. To optimize production processes and create higher-quality products, artificial intelligence technology has become the mainstream in the market. The development of robotic process automation provided the case company with a means for introducing new technologies in its corporate operations (e.g., business processes, labor management, supply chain management, procurement, and accounting) to improve corporate efficiency and reduce costs, thereby helping the company to make progress towards achieving sustainable development.

**Author Contributions:** S.Y.H.: Writing—review & supervision; A.A.C.: Writing—review & editing; P.C.C.: Writing—original draft; N.W.: Writing—Methodology.

**Funding:** This research received no external funding.

**Conflicts of Interest:** The authors declare no conflict of interest.

## Appendix A

**Table A1.** Taking QC2 as an example for Table 2.

| The calculation steps are as follows |
| --- |
| Material_ Resist 1: Standard Unit Price $3200 × Quantity of Usage $0.8 = Monetary Amount $2560 |
| Labor_Direct labor: Standard Unit Price $125 × Quantity of Usage $4 = Monetary Amount $500 |
| Labor_Indirect labor: Standard Unit Price $165 × Quantity of Usage $3 = Monetary Amount $495 |
| Manufacturing Costs_Depreciation expense: Standard Unit Price $10,000 × Quantity of Usage $30 = Monetary Amount $300,000 |
| Manufacturing Costs_Consumables: Standard Unit Price $800 × Quantity of Usage $4 = Monetary Amount $3200 |
| Manufacturing Costs_Other costs: Standard Unit Price $600 × Quantity of Usage $4 = Monetary Amount $2400 |
| Manufacturing_Electricity: Standard Unit Price $4500 × Quantity of Usage $4 = Monetary Amount $18,000 |
| The total cost of QC2 is $327,155 |

**Table A2.** Taking Tap Water's Cumulative Input Quantity and Cumulative Monetary Input as an example for Table 6.

| The calculation steps are as follows | | | |
| --- | --- | --- | --- |
| Step1 Monetary Input/Input Quantity = Average Unit Price Step2 last Cumulative Input Quantity + Input Quantity = Cumulative Input Quantity Step3 Cumulative Input Quantity × Average Unit Price = Cumulative Monetary Input | | | |
| Work Section | Step1 Average Unit Price | Step2 Cumulative Input Quantity | Step3 Cumulative Monetary Input |
| QC2 BM | $698,425/53,260 = $13.11 | 0 + 53,260 = 53,260 | 53,260 × $13.11 = $698,239 |
| QC3 BM | $2,916,395/222,398 = $13.11 | 53,260 + 222,398 = 275,658 | 275,658 × $13.11 = $3,613,876 |
| QC4 POL/ITO | $3,950,633/301,265 = $13.11 | 275,658 + 301,265 = 576,923 | 576,923 × $13.11 = $7,563,460 |
| QC5 PS | $4,590,404/350,053 = $13.11 | 576,923 + 350,053 = 926,976 | 926,976 × $13.11 = $12,152,655 |
| QC6 INS | $ 4,523,001/344,913 = $ 13.11 | 926,976 + 344,913 = 1,271,889 | 1,271,889 × $13.11 = $16,674,464 |
| Tap Water Monetary Input Subtotal: $ 698,425 + $,2,916,395 + 3,950,633 + $4,590,404 + $4,523,001 = 16,678,858 | | | |

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
