# Peer review of "The Application of Material Flow Cost Accounting in Waste Reduction"

_sustainability, doi:10.3390/su11051270_

Reviewer 1 Report

Comments on Manuscript ID: Sustainability-23048

Promoting Organizational Sustainable Development and the Application of Material Flow Cost Accounting in Waste Reduction

This paper deals with ISO 14051-based MFCA. The authors conducted a case study on a flat-panel parts supplier to determine whether efficient use of recycled glass could reduce company costs. I explain below why this paper is not ready for publication.

The paper is mainly unclear in terms of objectives. It seems a descriptive study and not an analytical case study. Case studies usually respond to ‘How’ and ‘Why’ questions. The research method is also not well explained. The paper is composed mainly by six (some long) tables that the authors believe are self-explanatory. In my view they are not. They should be introduced by the authors drawing attention to the main issues presented in the tables.

The way documents are referred is superficial. The documents are not well explained. For example, “In Taiwan, the government not only mandates publicly listed companies to publish financial reports but also requires several industries to produce CSR reports. An increasing number of industries have been included in the scope of application” (pp.1-2). This is mandated by which document? Several sentences later, p. 2, the authors argue “In 2007, the Environmental Protection Administration of Taiwan formulated a set of environmental-accounting guidelines, which are currently commonly adopted by Taiwanese corporations as the foundation for environmental accounting practices”. Is this document the same as the one cited several sentences before? Why not put everything together and explain in a clear way what the regulations on CSR are in Taiwan?

As mentioned before, this paper is about ISO 14051-based MFCA. But the authors seem to believe that the reader know the concepts well. They never explain the topic they are dealing with. This does not permit understanding of the main concepts of this paper.

On p. 4 the authors argue: “In traditional cost accounting, the basic calculation model involves allocating manufacturing costs to various products. Therefore, the value of wastes generated during the production process is “zero.” In the traditional literature in the topic, we can see that there are many ways to measure waste and defective products. The simplest are: zero cost or zero profit (in this case, the cost is the proceeds from the sale of waste and defective products). But there are companies which know exactly the cost associated with waste and defective products.

I could not find any contribution. What is new in this paper? The paper is mainly technical and descriptive. It does not extend knowledge. The results are not generalized. There is no theory in the paper that can be extended.

The research section needs improvement and a theory to help improve the contribution of the paper.

It was strange to see that in sections where authors deal with water and energy costs, the title of the section is always labelled as “Energy cost”.

P. 5: “Thus, in our case analysis, we first described the implementation manuals adopted in Taiwan and Japan and then analyzed the case company by following the implementation methods specified by the manual to circumvent the effects of subjectivity on this research.” I could not find the manuals. They seem likely to be important to understand the case study.

All the Tables and Figures need to be better explained. Table 8 is too long. I am not sure if all tables are necessary.

P.11: I do not agree with this sentence for the reason mentioned before: “Cost accounting incorporates the cost of losses into the cost of a finished product. For example, the quantity of consumption in Table 3 was recorded in terms of the quantity of losses on a work order form, but the cost of loss was “zero.” By contrast, MFCA involves the separate calculation of the sum of the cost of losses and determination of the causes for these losses. The focus of MFCA is on recognizing waste as nonmarketable products. In this study, we consolidated work orders for an entire year and listed all of the inputted materials, ascertaining “Input = positive products + negative products” to obtain a material flow model record sheet for Company T.” It can be true for company T. But the authors cannot generalize that the cost accounting systems of all companies allocate the cost of waste and defective materials to the finished products.

This sentence has something wrong: “Therefore, government promoting the implementation of MFCA should collect, analyze, compile, and describe data from the perspectives of financial and accounting specialists to alert these specialists to the difference between conventional cost accounting and MFCA. After a balance sheet for basic analysis has been obtained, it should then be administered to plant personnel for hotspot improvements.” It should be government to implement MFCA?

The conclusions are very poor. I would say that it was not necessary to do any study to conclude that “This study assists companies to examine the process of recycled glass production by using ISO 14051-based MFCA and investigate whether company can reduce costs by using recycled glass and mitigate the pollution and improve the companies’ environmental performances.” What are the empirical and theoretical implications of this study?

As acknowledged by the authors, the study has an important limitation: “The present study was not focused on the process of collecting used products.” (p.17). I would say that this is the most important issue in terms of waste management. The authors should at least explain why they did not address this.

Minor issues:

P. 1: what is the year of the WCED document?

P. 2, line 12: …on a voluntary basis.” This quote does not have a start? Additionally, the EU published a law that requires large companies to disclose certain information on the way they operate and manage social and environmental challenges. Thus, the information is not voluntary but required for large EU companies.

P. 3. This sentence is incomplete: “Elmualim [13] 71 used the construction industry as an example to discuss the relationship between CSR and sustainable development, indicating that under ideal circumstances."

P. 6: there are English problems in the bullet points (e.g. 2. “Environment-safety department: provide information… “.

The paper has too many abbreviations, making the reading very hard.

The referencing on p. 17 is not consistent with the protocol being used.

Author Response

Dear reviewer, thanks for providing us these good suggestions. Followings are our replies.

1. The paper is mainly unclear in terms of objectives. It seems a descriptive study and not an analytical case study. Case studies usually respond to ‘How’ and ‘Why’ questions. The research method is also not well explained.

Ans: Thanks for reviewer’s suggestion. We have added “Research method” section and try to explain the importance of these case study as follows:

Research method

The case study is one of the research methods in the social science. Case study method can disclosure the problem and then systematically collect and analyze the problem in a scientific way to find the solution (Yin 1994). This research adopted the case-study method using a flat-panel parts supplier as an example to analyze the implementation of ISO 14051-based MFCA by the case company and investigate the results. The case-study method is an empirical research approach that is used to comprehensively examine the realistic conditions of a case company. This method is most commonly used in exploratory research for which the relevant qualitative research techniques are convenient [26]. Although the case-study method provides a comprehensive understanding of a situation and causal relationships, it is still overly subjective and lacks objectivity. Thus, in our case analysis, we first described the implementation manuals adopted in Taiwan and and then analyzed the case company by following the implementation methods specified by the manual to circumvent the effects of subjectivity on this research.

This paper uses case study method to discuss the introduction of ISO 14051 MFCA. We need to collect the data of the actual cost for each quantity center and the work order to calculate the amount of material flow input and output from each quantity center. However, these are confidential financial and cost information within the company. Different companies have different cost data due to different process and information system construction. In the absence of database or cross-company consistency data, we use case studies to conduct empirical analysis. Some of management accounting related research (Banker et al., 2000; Gustafsson and Johnson, 2002; Gosman and Kohlbeck, 2009) use case study for the empirical analysis due to the lack of public information.

Although the case-study method provides a comprehensive understanding of a situation and causal relationships, it is still overly subjective and lacks objectivity. Thus, in our case analysis, we first described the implementation manuals adopted in Taiwan Industrial Development Bureau, Ministry of Economic Affairs then analyzed the case company by following the implementation methods specified by the manual to circumvent the effects of subjectivity on this research.

The financial information of the case company T is collected from the database of the Company T. The production process and cost structure are summarized from the database of company T. We also have in-depth interview with production manager, accounting manager and other internal personnel to further understand the flow of the company's existing product line and cost structure.

The importance of this case study is the process of applying MFCA can provide the suggestions for future installation of MFCA. For example, the setting of the quantity center should be the same as the existing data record and periodic review framework to avoid the ineffective use. Second, the historical data should be saved in the operating system, and the material flow currency should be quantitatively linked to the accounting, and the existing management system of each department should be comprehensively integrated to facilitate the rapid introduction of product. The analysis of material flow and energy use assists company understand the relationship between material usage and cost, identify the hot spots of material and cost loss, improve the efficiency of data analysis and systematically review the process to promote and strengthen the environmental cost management.

2.  The paper is composed mainly by six (some long) tables that the authors believe are self-explanatory. In my view they are not. They should be introduced by the authors drawing attention to the main issues presented in the tables.

Ans: Thanks for reviewer’s suggestion. The explanations of the Tables 1 to 7 are as follows:

Table 1. Description of each quantity center. 

Table 1 presents the process contents, input, output, and loss items of each quantity center. The input materials of each quantity center include direct materials, indirect materials, and all processed materials in a product. If auxiliary materials are excessively used and incurred wastewater-treatment costs, we advocated the incorporation of these materials in MFCA analysis.

Table 2. The actual cost for each quantity center.

The actual cost of each quantity center is summarized in Table 2.

Cost calculation in the ERP system began with the bill of material. Two types of work orders were issued: a standard-order BOM (realistic standard) and production-order BOM (ideal standard). The difference between them was used to adjust uncertain losses during the production and prepare for material distribution. The two types of work orders are described as follows:

(1)     Standard-order BOM: A standard-order BOM is the and also consists of the amount of materials used in each QC, hours of labor, and other manufacture quantity of input materials required to produce a piece of color filter without any substantial loss ring costs.

(2)     Production-order BOM: It is almost impossible that no loss incurred during the production process. Therefore, production-management staff adopt a standard-order BOM as the basis for adjusting input quantity according to the status of a production line. Adjustments are most commonly made to yields. On a production-order BOM, the quantity of usage is also the basis for calculating material stock and the quantity of materials to be purchased. Cost accounting is based on the data provided on a production-order BOM, which documents production start time, actual usage in each QC, and the amount of loss. In the case of actual usage, cost is calculated by multiplying BOM-based input quantities in each QC by the standard unit price. Standard unit price is the unit price of a most recently approved order. The actual cost of each QC is summarized in Table 2. In the following section, adjustments to figures in the table are simulated. After a work order has been completed and inventoried, a work order record sheet is completed, as displayed in Table 3.

Table 2. The actual cost for each QC.

Item

QC 1

QC 2

QC 3

QC 4

Total

Type

Name

Unit

Standard Unit Price

Quantity of

Usage

Monetary Amount

Quantity of Usage

Monetary Amount

Quantity of Usage

Monetary Amount

Quantity of Usage

Monetary Amount

Quantity of Usage

Monetary Amount

Material

Glass

Piece

  2,800

 303

848,400

-

-

     -    

303

   848,400

Resist 1

Kg

  3,200

-

 0.8

  2,560

-

     -    

0.8

     2,560

Resist 2

Kg

  1,800

 0.7

  1,260

-

-

     -    

0.7

     1,260

Resist 3

Kg

  1,500

-

-

-

 0.8

  1,200

0.8

     1,200

ITO

Kg

 21,000

-

-

 0.2

  4,200

-

0.2

     4,200

Labor

Direct labor

Hour

    125

 3.5

    438

   4

    500

   2

    250

 3.5

    438

13

     1,625

Indirect labor

Hour

    165

   3

    495

   3

    495

   3

    495

   3

    495

12

     1,980

Manufacturing Costs

Depreciation expense

Area/Square

 10,000

  10

100,000

  30

300,000

  20

200,000

  20

200,000

80

   800,000

Consumables

Hour

    800

 3.5

  2,800

   4

  3,200

   2

  1,600

 3.5

  2,800

13

    10,400

Other costs

Hour

    600

 3.5

  2,100

   4

  2,400

   2

  1,200

 3.5

  2,100

13

     7,800

Electricity

KWH

  4,500

 3.5

 15,750

   4

 18,000

   2

  9,000

 3.5

 15,750

13

    58,500

Total

971,243

327,155

216,745

222,783

 1,737,925

Taking QC2 as an example, the calculation steps are as follows:

Material_ Resist 1: Standard Unit Price $3,200 * Quantity of Usage $0.8=Monetary Amount $2,560

Labor_Direct labor: Standard Unit Price $125 * Quantity of Usage $4=Monetary Amount $500

Labor_Indirect labor: Standard Unit Price $165 * Quantity of Usage $3 =Monetary Amount $495

Manufacturing Costs_Depreciation expense: Standard Unit Price $10,000 * Quantity of Usage $30 =Monetary Amount $300,000

Manufacturing Costs_Consumables: Standard Unit Price $800 * Quantity of Usage $4 =Monetary Amount $3,200

Manufacturing Costs_Other costs: Standard Unit Price $600 * Quantity of Usage $4 =Monetary Amount $2,400

Manufacturing_Electricity: Standard Unit Price $4,500 * Quantity of Usage $4 =Monetary Amount $18,000

The total cost of QC2 is $327,155

Table 3. Quantity of materials used in a work order

The quantity of consumption in Table 3 was recorded in terms of the quantity of losses on a work order form, but the cost of loss was “zero.” By contrast, MFCA involves the separate calculation of the sum of the cost of losses and determination of the causes for these losses. In the following section, adjustments to figures in the table are simulated. After a work order has been completed and inventoried, a work order record sheet is completed, as displayed in Table 3.

Table 4. Scope of calculation

Material quantity is calculated using the total quantity of materials in a work order that has both input and output in the same year within the scope of the specified time period. Calculations were based on the scope of the calculation in Table 4.

Table 5. Energy cost calculation method

Company T selected the sixth generation plant primarily because the production line is equipped with unique water and electricity meters, which facilitated the collection of energy data. However, not all stations had an electricity meter. Therefore, we still used input quantity as the basis of the allocation. Table 5 presents details of the calculation method.

Table 6. Energy-cost calculation of Company T

Company T selected the sixth generation plant primarily because the production line is equipped with unique water and electricity meters, which facilitated the collection of energy data. However, not all stations had an electricity meter. Therefore, we still used input quantity as the basis of the allocation. Table 5 presents details of the calculation method.

Table 6. Energy-cost calculation of Company T

Unit: NT$

Item

Work Section

Input Quantity

Monetary Input

Average Unit Price

Cumulative Input Quantity

Cumulative Monetary Input

Tap Water

QC1  INC

QC2  BM

53,260

698,425

13.11

53,260

698,425

QC3  RGB

     222,398

     2,916,395

13.11

 275,658

3,614,820

QC4  POL/ITO

     301,265

      3,950,633

13.11

    576,923

7,565,453

QC5  PS

     350,053

    4,590,404

13.11

 926,976

12,155,857

QC6  INS

     344,913

     4,523,001

13.11

    1,271,889

16,678,858

Tap Water Subtotal

    1,271,889

 16,678,858

13.11

Electricity

QC1  INC

 2,905,791

     4,217,049

1.45

 2,905,791

     4,217,049

QC2  BM

 20,315,148

 29,482,497

1.45

 23,220,939

    33,699,546

QC3  RGB

 55,192,669

 80,098,737

1.45

 78,413,608

113,798,283

QC4  POL/ITO

 93,347,887

 135,471,756

1.45

171,761,495

 249,270,039

QC5  PS

108,138,933

 156,937,362

1.45

279,900,428

 406,207,401

QC6  INS

109,630,206

 159,101,582

1.45

389,530,634

 565,308,983

Electricity Subtotal

389,530,634

 565,308,983

1.45

Total

390,802,523

 581,987,841

1.49

Note: Figures adjusted through simulation

Taking Water cost as example:

Step1 Monetary Input / Input Quantity = Average Unit Price

Step2 last Cumulative Input Quantity + Input Quantity = Cumulative Input Quantity

Step3 Cumulative Input Quantity  * Average Unit Price = Cumulative Monetary Input

QC2  BM

Step 1

$698,425/53,260   =$ 13.11

Step 2

0 +   53,260 = 53,260

Step 3

53,260 * $13.11 =$ 698,239

QC3  BM

Step 1

$2,916,395/222,398   =$ 13.11

Step 2

53,260   + 222,398 = 275,658

Step 3

275,658 * $13.11 = $3,613,876

QC4    POL/ITO

Step 1

$3,950,633/301,265   =$ 13.11

Step 2

275,658   + 301,265 =576,923

Step 3

576,923 * $13.11 = $7,563,460

QC5  PS

Step 1

$   4,590,404/350,053 =$ 13.11

Step 2

576,923   + 350,053 = 926,976

Step 3

926,976 * $13.11 = $12,152,655

QC6    INS

Step 1

$   4,523,001/344,913 =$ 13.11

Step 2

926,976   + 344,913 = 1,271,889

Step 3

1,271,889 * $13.11 = $16,674,464

Tap Water Monetary Input Subtotal

$ 698,425 + $,2,916,395 +3,950,633+$4,590,404+$4,523,001=16,678,858

Table 7. System cost calculation method

System costs includes labor, depreciation expenses, consumables, and other expenses. With the exception of labor costs, system costs are fixed costs. Table 7 presents the system cost calculation method.

3. The way documents are referred is superficial. The documents are not well explained. For example, “In Taiwan, the government not only mandates publicly listed companies to publish financial reports but also requires several industries to produce CSR reports. An increasing number of industries have been included in the scope of application” (pp.1-2). This is mandated by which document? Several sentences later, p. 2, the authors argue “In 2007, the Environmental Protection Administration of Taiwan formulated a set of environmental-accounting guidelines, which are currently commonly adopted by Taiwanese corporations as the foundation for environmental accounting practices”. Is this document the same as the one cited several sentences before? Why not put everything together and explain in a clear way what the regulations on CSR are in Taiwan?

Ans: Thanks for reviewer’s suggestion. We have deleted the CSR document in our paper because the other reviewer suggests that CSR is not useful for the aim of the paper and the contribution. Conversely, we focus more on material, environmental and performance accounting, that is much more interesting considering your results.

Environment protection is critical for sustainability. Continuous investments in energy consumption and natural resource consumption as well as manufacturing sectors and infrastructure have had seriously harmful impacts on the environment (Mahmood, Furqan & Bagais, 2019). Environmental accounting create accountability for business entities in terms of their efforts to protect the environment in their corporate decisions (Masud, Bae, & Kim, 2017). Environmental accounting constitutes a tool for applying the sustainable development concept and now commands acceptance as a means of ensuring the preservation of the environment Environmental accounting information includes financial, environmental performance, and policy aspects, which are scattered in many parts of annual reports and social responsibility reports. The quality of disclosures can be characterized from aspects of comprehensiveness, reliability, and compliance (Masud, Bae, & Kim, 2017). In decision making, an organization considers different pressures from internal and external parties and attempts to legitimize the impact of its activities on the environment in the eyes of society and various pressure groups [8]. Environmental accounting plays an active role in preparing, presenting, and analyzing environmental information for interested parties holders, thus encouraging top management to improve environmental conditions (Masud, Bae, & Kim, 2017).

4. As mentioned before, this paper is about ISO 14051-based MFCA. But the authors seem to believe that the reader know the concepts well. They never explain the topic they are dealing with. This does not permit understanding of the main concepts of this paper.

Ans: Thanks for reviewer’s suggestion. We have added the introduction of ISO 14501 MFCA as follows:

ISO 14051 defines MFCA as a “tool for quantifying the flows and stocks of materials in processes or production lines in both physical and monetary units” where ‘materials’ include energy and water (ISO, 2011, p. 3). These flows and stocks of materials are important because they pervade business practice (Jasch, 2006). The aim of MFCA is to provide information to management about opportunities for reducing materials use and improving monetary performance of businesses at the same time, an irresistible opportunity.

To facilitate implementation, ISO 14051 proposes several MFCA implementation steps as delineated below. The level of detail and complexity of the analysis will depend on the size of the organization, nature of the organization’s activities and products, number of processes, and quantity centers chosen for analysis, among other factors. These conditions make MFCA a flexible tool that can be applied in a wide range of organizations.

Ø   IMPLEMENTATION STEP 1: ENGAGING MANAGEMENT AND DETERMINING ROLES AND RESPONSIBILITIES (Clause 6.2, 6.3, ISO 14051:2011)

Successful projects usually start with support from the company’s management; MFCA is not an exception. If the company management understands the benefits of MFCA and its usefulness in achieving an organization’s environmental and financial targets, it is easier to gain commitment from the whole organization. In order to be effectively implemented, it is highly recommended that top management take the lead in MFCA implementation by assigning roles and responsibilities, including setting up an MFCA project implementation team, providing resources, monitoring progress, reviewing results, and deciding on improvement measures based on the MFCA results.

In general terms, management should be engaged in all phases of MFCA implementation and it is recommended that MFCA projects start with aggressive support from management, followed by a bottom-up approach on-site.

In addition, successful implementation of MFCA requires collaboration between different departments within the organization. The reason why collaboration is needed is because different sources of information are required to complete MFCA analysis. Through engagement of the company management in the MFCA implementation process, the required expertise can be determined and the correct flow of information between all areas involved can be facilitated.

Ø   IMPLEMENTATION STEP 2: SCOPE AND BOUNDARY OF THE PROCESS AND ESTABLISHING A MATERIAL FLOW MODEL (Clauses 6.4, 6.5, ISO 14051:2011)

Based on collected material flow data, the MFCA boundary needs to be specified to understand clearly the scale of MFCA activity. During implementation, it is usually recommended to focus on specific products or processes at the beginning and then expanding implementation to other products. By implementing MFCA in steps, the analysis is simplified and better results can be achieved.

The boundary can be limited to a single process, multiple processes, an entire facility, or a supply chain. It is recommended that the process or processes that are chosen for initial implementation be the ones with potentially significant environmental and economic impacts. After specifying the boundary, the process should be classified in quantity centers using process information and procurement records. In MFCA, the quantity center is the part of the process in which inputs and outputs are quantified. In most cases, quantity centers represent parts of the process in which materials are transformed. If the material flow between two processes is the source of significant material loss, the flow can be classified as a separate material flow.

After determining the boundary and quantity centers, a time period for MFCA data collection needs to be specified. While MFCA does not indicate the period during which data must be collected for analysis, it should be sufficiently long to allow meaningful data to be collected and to minimize the impact of any significant process variation that can affect the reliability and usability of the data, such as seasonal fluctuations. Several historical MFCA projects indicate that the appropriate data collection period can be as short as a month, with a half-year or a year of data collection being the most common.

Ø   IMPLEMENTATION STEP 3: COST ALLOCATION (Clauses 5.3, 6.8, ISO 14051:2011)

MFCA divides costs into the following categories:

Material cost: cost for a substance that enters and/or leaves a quantity center

Energy cost: cost for electricity, fuel, steam, heat, and compressed air

System cost: Cost of labor, cost of depreciation and maintenance, and cost of transport

Waste management cost: cost of handling waste generated in a quantity center

Material costs, energy costs, and system costs are assigned or allocated to either products or material losses at each quantity center based on the proportion of the material input that flows into product and material loss. The material costs for each input and output flow are quantified by multiplying the physical amount of the material flow by the unit cost of the material over the time period chosen for the analysis. When quantifying the material costs for products and material losses, the material costs associated with any changes in material inventory within the quantity center should also be quantified. In contrast to material, energy, and system costs that are assigned to products and material losses proportionally, 100% of the waste management costs are attributed to material loss, since the costs represent the costs of managing this material loss.

Ø   IMPLEMENTATION STEP 4: INTERPRETING AND COMMUNICATING MFCA RESULTS (Clauses 6.9, 6.10, ISO 14051:2011)

MFCA implementation provides information such as material loss throughout the process, the use of materials that do not become products, overall costs, and energy and system costs associated with the material loss. This information brings about multiple impacts by increasing the awareness of the company’s operations. Managers who are aware of the costs associated with material losses can then identify opportunities to increase efficiency in material use and improve business performance.

Ø   IMPLEMENTATION STEP 5: IMPROVING PRODUCTION PRACTICES AND REDUCING MATERIAL LOSS THROUGH MFCA RESULTS (Clause 6.11, ISO 14051:2011)

Once MFCA analysis has assisted an organization to understand the magnitude, consequences, and drivers of material use and loss, the organization may review the MFCA data and seek opportunities to improve environmental and financial performance. The measures taken to achieve these improvements can include substitution of materials; modification of processes, production lines, or products; and intensified research and development activities related to material and energy efficiency.

5.On p. 4 the authors argue: “In traditional cost accounting, the basic calculation model involves allocating manufacturing costs to various products. Therefore, the value of wastes generated during the production process is “zero.” In the traditional literature in the topic, we can see that there are many ways to measure waste and defective products. The simplest are: zero cost or zero profit (in this case, the cost is the proceeds from the sale of waste and defective products). But there are companies which know exactly the cost associated with waste and defective products.

Ans: Thanks for reviewer’s suggestion. We have revised the paragraph as follows:

In traditional cost accounting, the basic calculation model involves allocating manufacturing costs to various products. Therefore, the value of wastes generated in most companies during the production process is “zero.”

6. I could not find any contribution. What is new in this paper? The paper is mainly technical and descriptive. It does not extend knowledge. The results are not generalized. There is no theory in the paper that can be extended.

Ans: Thanks for reviewer’s suggestion. We have added the contributions of the paper as follows:

The study promotes the application of MFCA to SMEs in Malaysia because the case company sets an example to other SMEs that MFCA can be embarked on with relative ease, and indicates perhaps an inexpensive cost of implementation

The practical contributions of the study are as follows:

This study analyzes Taiwan CFI Company, the first company to receive ISO 14051 MFCA certification in Taiwan, which is recommended to get the business environmental protection award. The case company can be the role model of promoting the business sustainability. The experience of applying MFCA can be the guidelines of the business in manufacturing and service industries, which are devoted to improve the sustainable environment and lower the costs.

The MFCA analysis shows that the recycling process increase reprocessing cost which is not beneficial for case company to promote the sustainability. It provides case company the improvement directions and guidelines for the waste recycling after the application of MFCA. The results can utilize Material Circularity Indicator, developed by Ellen MacArthur Foundation, to analyze the actual recovery rate of the materials and the life or frequency of the product use, which can be the reference for the improved design of the second-hand recycling process system.

7. It was strange to see that in sections where authors deal with water and energy costs, the title of the section is always labelled as “Energy cost”.

Ans: Thanks for reviewer’s suggestion. In this paper, Energy cost includes water and electricity.

8.P. 5: “Thus, in our case analysis, we first described the implementation manuals adopted in Taiwan and Japan and then analyzed the case company by following the implementation methods specified by the manual to circumvent the effects of subjectivity on this research.” I could not find the manuals. They seem likely to be important to understand the case study.

Ans: Thanks for reviewer’s suggestion. We have added the manual as attachment. <+manual>

https://www.idbcfp.org.tw/DownloadDetail.aspx?id=5

9.All the Tables and Figures need to be better explained. Table 8 is too long. I am not sure if all tables are necessary.

Ans: Thanks for reviewer’s suggestion. We have deleted Table 8.

10. P.11: I do not agree with this sentence for the reason mentioned before: “Cost accounting incorporates the cost of losses into the cost of a finished product. For example, the quantity of consumption in Table 3 was recorded in terms of the quantity of losses on a work order form, but the cost of loss was “zero.” By contrast, MFCA involves the separate calculation of the sum of the cost of losses and determination of the causes for these losses. The focus of MFCA is on recognizing waste as nonmarketable products. In this study, we consolidated work orders for an entire year and listed all of the inputted materials, ascertaining “Input = positive products + negative products” to obtain a material flow model record sheet for Company T.” It can be true for company T. But the authors cannot generalize that the cost accounting systems of all companies allocate the cost of waste and defective materials to the finished products.

Ans: Thanks for reviewer’s suggestion. We have revise the paragraph as follows:

Cost accounting of company T incorporates the cost of losses into the cost of a finished product. For example, the quantity of consumption in Table 3 was recorded in terms of the quantity of losses on a work order form, but the cost of loss was “zero.” By contrast, MFCA involves the separate calculation of the sum of the cost of losses and determination of the causes for these losses. The focus of MFCA is on recognizing waste as nonmarketable products. In this study, we consolidated work orders for an entire year and listed all of the inputted materials, ascertaining “Input = positive products + negative products” to obtain a material flow model record sheet for Company T.”

11. This sentence has something wrong: “Therefore, government promoting the implementation of MFCA should collect, analyze, compile, and describe data from the perspectives of financial and accounting specialists to alert these specialists to the difference between conventional cost accounting and MFCA. After a balance sheet for basic analysis has been obtained, it should then be administered to plant personnel for hotspot improvements.” It should be government to implement MFCA?

Ans: Thanks for reviewer’s suggestion. We have revised the sentence as follows:

Therefore, the implementation of MFCA should collect, analyze, compile, and describe data from the perspectives of financial and accounting specialists to alert these specialists to the difference between conventional cost accounting and MFCA. After a balance sheet for basic analysis has been obtained, it should then be administered to plant personnel for hotspot improvements.

12. The conclusions are very poor. I would say that it was not necessary to do any study to conclude that “This study assists companies to examine the process of recycled glass production by using ISO 14051-based MFCA and investigate whether company can reduce costs by using recycled glass and mitigate the pollution and improve the companies’ environmental performances.” What are the empirical and theoretical implications of this study?

Ans:  Thanks for the reviewer’s suggestion. We have added the empirical and theoretical implications of this study.

The study promotes the application of MFCA to SMEs in Taiwan because the case company sets an example to other SMEs that MFCA can be embarked on with relative ease, and indicates perhaps an inexpensive cost of implementation

The MFCA analysis shows that the recycling process increase reprocessing cost which is not beneficial for case company to promote the sustainability. It provides case company the improvement directions and guidelines for the waste recycling after the application of MFCA. The results can utilize Material Circularity Indicator, developed by Ellen MacArthur Foundation, to analyze the actual recovery rate of the materials and the life or frequency of the product use, which can be the reference for the improved design of the second-hand recycling process system.

13. As acknowledged by the authors, the study has an important limitation: “The present study was not focused on the process of collecting used products.” (p.17). I would say that this is the most important issue in terms of waste management. The authors should at least explain why they did not address this.

Ans: Thanks for reviewer’s suggestion. The explanation is as follows:

The reliability of recycled glass in the recovery process should be evaluated before the study. This study does not focus on the collection process of used products. The evaluation of the effectiveness and reliability of the recycling process requires further steps, which can be carried out by manufacturers, retailers or third parties. That is much more suitable for the follow-up research.

Minor issues:

P. 1: what is the year of the WCED document?

Ans: Thanks for reviewer’s opinion. WCED document is in year 1987.

P. 2, line 12: …on a voluntary basis.” This quote does not have a start? Additionally, the EU published a law that requires large companies to disclose certain information on the way they operate and manage social and environmental challenges. Thus, the information is not voluntary but required for large EU companies.

Ans: Thanks for reviewer’s suggestion. We have revised the sentence as follows:

The European Union defined CSR as a concept whereby companies are required to integrate social and environmental concerns in their business operations and in their interaction with their stakeholders.

P. 3. This sentence is incomplete: “Elmualim [13] 71 used the construction industry as an example to discuss the relationship between CSR and sustainable development, indicating that under ideal circumstances."

Ans: Thanks for reviewer’s opinion. We have revised the sentence as follows:

Elmualim [13] used the construction industry as an example to discuss the relationship between CSR and sustainable development.  The result indicated that under ideal circumstances, CSR policy functions as an integral, self-regulating mechanism in a business model in which a business monitors and ensures the adherence to the law and ethical standards.

P. 6: there are English problems in the bullet points (e.g. 2. “Environment-safety department: provide information… “.

Ans: Thanks for reviewer’s suggestion. We have revised the part as follows:

The responsibilities of each department are as follows:

1.      Finance department integrates the construction and cost information to construct a material flow model, compile reports, and document findings for future reference. It also provides valuable assistance in the implementation process, from preliminary planning to the compilation of an MFCA balance sheet. In the preparatory stage of MFCA implementation, a scope and time period must first be specified.

2.      Environmental-safety department provides the information regarding the environmental effects of wastes, including types of the waste and waste-treatment activities.

3.      Plant affairs department identifies types and methods of the energy application and quantifies the consumption of the energy, including the energy loss in each quantity center (QC).

4.      Manufacturing and technical department analyzes the material balance in a material flow model and assists in providing the missing information.

The paper has too many abbreviations, making the reading very hard.

Ans: Thanks for reviewer’s suggestion. We have delete most of the abbreviations to make the paper much more readable.   

Reviewer 2 Report

Dear Authors,

Thank you for giving me the possibility to review your paper that is presenting an interesting and new topic in the field of material accounting and measurement. This research would have a scholar relevance, but probably at a further stage of development.

At this point, the article must be rewritten in accordance with the journal style and the methodology of contextual analysis. I suggest you enlarge the literature on your issue and to pay more attention to the methodology section that is not available in the current version of the paper. Also, the results section is very long and need to be better organized. These adjustments will help readers and reviewers in following the flow of your paper.

Please see my further comments below.

General comments

Title: is not clear how do you “Promoting Organizational Sustainable Development” rather it is clear that you are studying the “Application of Material Flow Cost Accounting in Waste Reduction”. I suggest you delete the first part.

Keywords- Abstract: avoid acronyms in the keywords and abstract

Introduction

The overall question on CSR seems not useful for the aim of the paper and your contribution. Delete From abstract, introduction, literature e conclusion this aspect and you will see that it will decrease the value of your research.

Conversely, your focus seems material, environmental and performance accounting, that is much more interesting considering your results.

The ISO 14051 must be introduced both in the introduction and in an appropriate section in the literature. Its link with the MFCA is not clear in the introduction.

Contribution seems clear but is not supported by a literature gap and a research aim. A research question will help you to create your research framework and related methodology. The latter is my main issue in the paper, that does not have a methodology section.

Here—and in your methodology as well— explain why you are employing the case study methodology, why did you chose that case and why in the Taiwan context. How you collect the data is also obscure. Studying Sustainability journal you will find a number of well-developed methodologies that embed the case study framework.

Name you case as Company T from the beginning, “the case company” is not so clear to understand into the text.

According to my previous comment delete section 2.1.

Section 2.2 and 2.3: Dedicate more attention on material flow management, considering also the management accounting literature (cost-accounting system) that you are going to mention in the comparison through the case study analysis. Also, a subsection on the ISO 14051 should be added.

147-170: these paragraphs should be moved into the introduction or in the methodology section.

Methodology: is missing

Please provide a methodology section. It seems you are employing a case study, but it is not presented until now: How, when and what data you have collected from this case?

The results section is quite long in its length and must be better organized and focused because the flow of the discourse is difficult to follow.

Figure 1 Is not readable.

Linea 175: what do you mean with "strategically collaborate": Please reframe the sentence.

185-190: reframe this sentence. It is not clear.

Discussion: is missing

Conclusions

At the moment, conclusions are just a summary of the previous sections on results. Discussion and conclusions are not supported by the previous studies.

This study presents a story with potential but not at this stage of the paper’s development. I strongly recommend that you focus first on define a research question then write a methodology. Results have to be strongly revised to render the flow and readable.

I hope my comments will help you.

Author Response

Dear reviewer, thanks for providing us these good suggestions. Followings are our replies.

1.General comments Title: is not clear how do you “Promoting Organizational Sustainable Development” rather it is clear that you are studying the “Application of Material Flow Cost Accounting in Waste Reduction”. I suggest you delete the first part.

Ans: Thanks for the reviewer’s suggestions. We have revised the title as “The Application of Material Flow Cost Accounting in Waste Reduction”

2. Keywords- Abstract: avoid acronyms in the keywords and abstract

Ans: Thanks for the reviewer’s suggestions. We have revised the Abstract and Keyword as follows:

Abstract: Due to the rise in environmental awareness, corporate companies have shifted their focus from an obsession with short-term profits to contemplating long-term strategies to achieve sustainable management. Effective use of resources is the primary indicator of this achievement. Fulfillment of corporate social responsibility and thinking beyond the regulatory aspects of corporate sustainable management are goals that have continually attracted attention worldwide. Material flow cost accounting based on ISO 14051, which was announced by the International Organization for Standardization (ISO), is a tool that can be used to achieve a balance between the environment and economy. We focused on using ISO 14051-based material flow cost accounting as an analytical evaluation tool from the perspectives of finance and accounting personnel. We conducted a case study on a flat-panel parts supplier to determine whether efficient use of recycled glass could reduce company costs. Our primary finding: the film layer on recycled washed glass tends to be stripped during the production process, causing increased reprocessing costs and thus rendering the cost of renewable cleaning higher than that of reworking. This study revealed that the ISO 14051-based material flow cost accounting analysis constitutes a valuable management tool, thereby facilitating the promotion of sustainable development.

Keywords: sustainability; corporate social responsibility; ISO14051; material flow cost accounting

3.  Introduction The overall question on CSR seems not useful for the aim of the paper and your contribution. Delete from abstract, introduction, literature e conclusion this aspect and you will see that it will decrease the value of your research. Conversely, your focus seems material, environmental and performance accounting, that is much more interesting considering your results.

Ans: Thanks for reviewer’s suggestion. We have deleted the CSR documents from this paper and added parts discussing environmental and performance accounting as follows:

Environment protection is critical for sustainability. Continuous investments in energy consumption and natural resource consumption as well as manufacturing sectors and infrastructure have had seriously harmful impacts on the environment (Mahmood, Furqan & Bagais, 2019). Environmental accounting create accountability for business entities in terms of their efforts to protect the environment in their corporate decisions (Masud, Bae, & Kim, 2017). Environmental accounting constitutes a tool for applying the sustainable development concept and now commands acceptance as a means of ensuring the preservation of the environment Environmental accounting information includes financial, environmental performance, and policy aspects, which are scattered in many parts of annual reports and social responsibility reports. The quality of disclosures can be characterized from aspects of comprehensiveness, reliability, and compliance (Masud, Bae, & Kim, 2017). In decision making, an organization considers different pressures from internal and external parties and attempts to legitimize the impact of its activities on the environment in the eyes of society and various pressure groups. Environmental accounting plays an active role in preparing, presenting, and analyzing environmental information for interested parties holders, thus encouraging top management to improve environmental conditions (Masud, Bae, & Kim, 2017).

4. The ISO 14051 must be introduced both in the introduction and in an appropriate section in the literature. Its link with the MFCA is not clear in the introduction.

Ans: Thanks for reviewer’s suggestions. We have added ISO14051 in the Introduction, Literature review and Research method.

Introduction:

To standardize MFCA practices, a working group of the ISO Technical Committee ISO/ TC 207, Environmental Management, developed ISO 14051, which complements the ISO 14000 family of environmental management system standards, including life cycle assessment (ISO 14040, ISO 14044) and environmental performance evaluation (ISO 14031). The standard was published in the second half of 2011.

Material flow cost accounting (MFCA) based on ISO 14051 is an environmental-management accounting (EMA) tool providing the analysis of environmental and economic balance [3]. ISO 14051 provides a general framework for material flow cost accounting.

Literature review

ISO 14051 defines MFCA as a “tool for quantifying the flows and stocks of materials in processes or production lines in both physical and monetary units” where ‘materials’ include energy and water (ISO, 2011, p. 3). These flows and stocks of materials are important because they pervade business practice (Jasch, 2006). The aim of MFCA is to provide information to management about opportunities for reducing materials use and improving monetary performance of businesses at the same time, an irresistible opportunity.

To facilitate implementation, ISO 14051 proposes several MFCA implementation steps as delineated below. The level of detail and complexity of the analysis will depend on the size of the organization, nature of the organization’s activities and products, number of processes, and quantity centers chosen for analysis, among other factors. These conditions make MFCA a flexible tool that can be applied in a wide range of organizations.

Ø   IMPLEMENTATION STEP 1: ENGAGING MANAGEMENT AND DETERMINING ROLES AND RESPONSIBILITIES (Clause 6.2, 6.3, ISO 14051:2011)

Successful projects usually start with support from the company’s management; MFCA is not an exception. If the company management understands the benefits of MFCA and its usefulness in achieving an organization’s environmental and financial targets, it is easier to gain commitment from the whole organization. In order to be effectively implemented, it is highly recommended that top management take the lead in MFCA implementation by assigning roles and responsibilities, including setting up an MFCA project implementation team, providing resources, monitoring progress, reviewing results, and deciding on improvement measures based on the MFCA results.

In general terms, management should be engaged in all phases of MFCA implementation and it is recommended that MFCA projects start with aggressive support from management, followed by a bottom-up approach on-site.

In addition, successful implementation of MFCA requires collaboration between different departments within the organization. The reason why collaboration is needed is because different sources of information are required to complete MFCA analysis. Through engagement of the company management in the MFCA implementation process, the required expertise can be determined and the correct flow of information between all areas involved can be facilitated.

Ø   IMPLEMENTATION STEP 2: SCOPE AND BOUNDARY OF THE PROCESS AND ESTABLISHING A MATERIAL FLOW MODEL (Clauses 6.4, 6.5, ISO 14051:2011)

Based on collected material flow data, the MFCA boundary needs to be specified to understand clearly the scale of MFCA activity. During implementation, it is usually recommended to focus on specific products or processes at the beginning and then expanding implementation to other products. By implementing MFCA in steps, the analysis is simplified and better results can be achieved.

The boundary can be limited to a single process, multiple processes, an entire facility, or a supply chain. It is recommended that the process or processes that are chosen for initial implementation be the ones with potentially significant environmental and economic impacts. After specifying the boundary, the process should be classified in quantity centers using process information and procurement records. In MFCA, the quantity center is the part of the process in which inputs and outputs are quantified. In most cases, quantity centers represent parts of the process in which materials are transformed. If the material flow between two processes is the source of significant material loss, the flow can be classified as a separate material flow.

After determining the boundary and quantity centers, a time period for MFCA data collection needs to be specified. While MFCA does not indicate the period during which data must be collected for analysis, it should be sufficiently long to allow meaningful data to be collected and to minimize the impact of any significant process variation that can affect the reliability and usability of the data, such as seasonal fluctuations. Several historical MFCA projects indicate that the appropriate data collection period can be as short as a month, with a half-year or a year of data collection being the most common.

Ø   IMPLEMENTATION STEP 3: COST ALLOCATION (Clauses 5.3, 6.8, ISO 14051:2011)

MFCA divides costs into the following categories:

Material cost: cost for a substance that enters and/or leaves a quantity center

Energy cost: cost for electricity, fuel, steam, heat, and compressed air

System cost: Cost of labor, cost of depreciation and maintenance, and cost of transport

Waste management cost: cost of handling waste generated in a quantity center

Material costs, energy costs, and system costs are assigned or allocated to either products or material losses at each quantity center based on the proportion of the material input that flows into product and material loss. The material costs for each input and output flow are quantified by multiplying the physical amount of the material flow by the unit cost of the material over the time period chosen for the analysis. When quantifying the material costs for products and material losses, the material costs associated with any changes in material inventory within the quantity center should also be quantified. In contrast to material, energy, and system costs that are assigned to products and material losses proportionally, 100% of the waste management costs are attributed to material loss, since the costs represent the costs of managing this material loss.

Ø   IMPLEMENTATION STEP 4: INTERPRETING AND COMMUNICATING MFCA RESULTS (Clauses 6.9, 6.10, ISO 14051:2011)

MFCA implementation provides information such as material loss throughout the process, the use of materials that do not become products, overall costs, and energy and system costs associated with the material loss. This information brings about multiple impacts by increasing the awareness of the company’s operations. Managers who are aware of the costs associated with material losses can then identify opportunities to increase efficiency in material use and improve business performance.

Ø   IMPLEMENTATION STEP 5: IMPROVING PRODUCTION PRACTICES AND REDUCING MATERIAL LOSS THROUGH MFCA RESULTS (Clause 6.11, ISO 14051:2011)

Once MFCA analysis has assisted an organization to understand the magnitude, consequences, and drivers of material use and loss, the organization may review the MFCA data and seek opportunities to improve environmental and financial performance. The measures taken to achieve these improvements can include substitution of materials; modification of processes, production lines, or products; and intensified research and development activities related to material and energy efficiency.

5.  Contribution seems clear but is not supported by a literature gap and a research aim. A research question will help you to create your research framework and related methodology. The latter is my main issue in the paper, that does not have a methodology section. Here—and in your methodology as well— explain why you are employing the case study methodology, why did you chose that case and why in the Taiwan context. How you collect the data is also obscure.

Ans: Thanks for reviewer’s suggestions. We have added The “Research method” section

Research method

The case study is one of the research methods in the social science. Case study method can disclosure the problem and then systematically collect and analyze the problem in a scientific way to find the solution (Yin 1994). This research adopted the case-study method using a flat-panel parts supplier as an example to analyze the implementation of ISO 14051-based MFCA by the case company and investigate the results. The case-study method is an empirical research approach that is used to comprehensively examine the realistic conditions of a case company. This method is most commonly used in exploratory research for which the relevant qualitative research techniques are convenient [26]. Although the case-study method provides a comprehensive understanding of a situation and causal relationships, it is still overly subjective and lacks objectivity. Thus, in our case analysis, we first described the implementation manuals adopted in Taiwan and and then analyzed the case company by following the implementation methods specified by the manual to circumvent the effects of subjectivity on this research.

This paper uses case study method to discuss the introduction of ISO 14051 MFCA. We need to collect the data of the actual cost for each quantity center and the work order to calculate the amount of material flow input and output from each quantity center. However, these are confidential financial and cost information within the company. Different companies have different cost data due to different process and information system construction. In the absence of database or cross-company consistency data, we use case studies to conduct empirical analysis. Some of management accounting related research (Banker et al., 2000; Gustafsson and Johnson, 2002; Gosman and Kohlbeck, 2009) use case study for the empirical analysis due to the lack of public information.

Although the case-study method provides a comprehensive understanding of a situation and causal relationships, it is still overly subjective and lacks objectivity. Thus, in our case analysis, we first described the implementation manuals adopted in Taiwan Industrial Development Bureau, Ministry of Economic Affairs then analyzed the case company by following the implementation methods specified by the manual to circumvent the effects of subjectivity on this research.

The financial information of the case company T is collected from the database of the Company T. The production process and cost structure are summarized from the database of company T. We also have in-depth interview with production manager, accounting manager and other internal personnel to further understand the flow of the company's existing product line and cost structure.

The importance of this case study is the process of applying MFCA can provide the suggestions for future installation of MFCA. For example, the setting of the quantity center should be the same as the existing data record and periodic review framework to avoid the ineffective use. Second, the historical data should be saved in the operating system, and the material flow currency should be quantitatively linked to the accounting, and the existing management system of each department should be comprehensively integrated to facilitate the rapid introduction of product. The analysis of material flow and energy use assists company understand the relationship between material usage and cost, identify the hot spots of material and cost loss, improve the efficiency of data analysis and systematically review the process to promote and strengthen the environmental cost management.

6.  According to my previous comment delete section 2.1. Section 2.2 and 2.3: Dedicate more attention on material flow management, considering also the management accounting literature (cost-accounting system) that you are going to mention in the comparison through the case study analysis.

Ans: Thanks for reviewer’s suggestion. We have added the literature of material flow management and cost accounting system

Material Flow Management

Adopting CSR and sustainable development while also retaining business competitiveness is a common goal in corporate communities. Because of this trend, a goal-oriented innovative management method is necessary for economic optimization and the reduction of material-related environmental pollution. Wagner and Enzler [20] explained that material flow management (MFM) which contributes to full utilization of the potential sustainable management of a company, provides a new framework for economic research, and initiates standardization processes for sustainable management. Because material flow and its effects are direct causes of ecological problems, MFM can be used to directly address the root of a problem and facilitate the reduction in the environmental pollution, which also leads to cost reduction [21].

        MFM is focused on optimizing systems rather than a single product or material and is primarily characterized by interdisciplinarity and networking on the basis of a normative orientation [20]. The initial step of MFM is goal setting. Because MFM is a component of a superordinate management concept, the goals of MFM arise from the normative and strategic goals of the company. In fact, the environmental-management tool with life-cycle-assessment orientation is designed to reveal the life-cycle–related effects of products and services on the natural environment. This approach helps to reduce environmental damage but does not provide clear contributions to cost savings or corporate profits [21]. Traditional cost-accounting methods in the evaluation of material flow processes neglect the complexity of material and energy flow [21]. Environmental-management accounting has expanded rapidly, and MFCA has become one of the most promising environmental-management accounting tools [22].

MFCA is an environmental-management accounting practice proposed by Professor Wagner from the Institut fur Management und Umwelt in Augsburg. MFCA aims to reduce material input through analysis. All input materials equal the amount of finished goods (positive products) plus that of generated waste (negative products). This equation represents the identification of material balance. When the number of positive products is constant, the number of negative products decreases, which reduces the environmental effect and amount of waste produced.

Nakajima [23] indicated that the production process in traditional cost accounting is a consumption process of economic value in which the added value of a product is valuated, and the consumption in an official production process is regarded as cost accounting. All monetary values of inputted resources are calculated in the production cost of a product as the output of the corresponding production process, meaning that a product should be burdened with all of the costs of utilized resources. Additionally, waste is not recognized in traditional cost accounting because it is considered to be unrelated to a value chain of various inputted resources (monetary value and physical volume), which are completely separated from production processes. By contrast, in MFCA analysis, wastes are not regarded as a process of value addition. Unlike traditional cost accounting, MFCA aims to recover the value of a product in a production process by equally valuing all of the outputs from a production process [23].

When using traditional cost-accounting methods, management authorities are most focused on the costs of manufacturing a product, which cover all of the costs incurred during the production process, including direct materials, labor, and production overheads. All of the costs incurred when manufacturing a product comprise the cost of the product. Irrespective of the amount of losses generated in an output process, the cost of losses is still allocated to product cost.

In MFCA, quantity of input refers to the actual quantity of input remaining in the final product. Input in MFCA differs from input in a manufacturing and production process in that the quantity of input in the final product is the actual amount of input required to produce 100% yield without taking yield into consideration. Yield is generally considered in the production process, producing a standard value, which represents a standard dosage that is considered to be the target value. At the beginning of the production process, quantity of allowable losses in management is considered in addition to yield, and yield plus the quantity of allowable losses leads to the development of an ideal standard. The difference between a realistic standard and ideal standard is the loss examined in MFCA; accordingly, the hidden value in this loss can be identified. 

In MFCA, waste is considered to be negative products. The main consequence of this assumption is that the manufacturing cost is used to produce not only the required products but also the undesired negative products (waste). Thus, waste is considered to bear part of the processing cost of all upstream processing steps. Conventional cost-accounting practices typically overlook significant cost incurred by waste; therefore, the cost incurred by waste is known as hidden cost. To reduce this cost and thereby improve economic performance, waste recovery is an essential strategy for manufacturing companies (Yoke Kin Wan, Rex T. L. Ng, Denny K. S. Ng,  Raymond, 2015).

Numerous studies worldwide have identified the advantages of MFCA. Based on a case study of a Sri Lankan crepe rubber factory, Dunuwila et al. (2018) adopted a cost-efficient and eco-friendly approach to improve the processing sector for natural rubber in the following three phases. 1. Quantifying factory resource use, economic loss, and greenhouse gas emissions through material flow analysis, MFCA, and life-cycle assessment [M W1] 2. developing proposals for viable improvement options; and 3. confirming the benefits of implementing the suggested improvement options. The results indicated that the proposed methods enabled 26% and 79% decreases in cost and global warming potential,[VS2]  respectively. Provided that the improvement options are viable, the results published by Dunuwila et al. (2018) can be used to ultimately increase profits for the rubber company, reduce the amount of toxic gas released into the atmosphere, reduce water consumption, and enhance the company image. Kokubu and Kitada (2015) indicated that intensely competitive Japanese companies have identified further room for improvement by using MFCA because the loss concept in MFCA is different from the general concept of compliance in traditional business management. The value of MFCA-derived information is attributed to this concept of difference in loss.

Nitto is a leading Japanese diversified materials manufacturer that offers a wide variety of products, including tapes, vinyl, LCDs, insulation, and reverse osmosis membranes. With the support of the Japanese Ministry of Economy, Nitto was the first model company to use MFCA. Before introducing MFCA, Nitto has long been committed to enhancing the efficiency and mitigating the environmental effects of its operations. Nitto discovered that its environmental-accounting system was incomplete primarily because only environmental protection costs were considered, and a few costs associated with environmental effects were still hidden. The environmental protection costs incurred through conventional environmental accounting only represented 1.6% of total sales in 2005. However, other costs, such as energy-consumption costs related to environmental effects are usually regarded as production costs, which accounted for approximately 13.6% of total sales, substantially higher than that for environmental protection costs (1.6%). During the process of target-product selection, Nitto focused on a product that exhibited an upward market trend and had production lines that generated a considerable amount of material loss and consumed a substantial amount of energy during the manufacturing process. Through MFCA, Nitto could accurately assess its material losses and associated costs. This information became a driving force for Nitto to improve the material productivity of this production line. Hence, the implementation team established its improvement plans on the basis of a cost–benefit analysis using the physical and cost data under MFCA. The plans and countermeasures were subsequently planned and implemented. Nitto recognized MFCA as a method that was helpful for clarifying losses in terms of cost, enabled managers to set explicit improvement goals, and provided a means for identifying potential cost reductions and positive effects on the environment. Using a professional management method such as MFCA ensured increased competitiveness for achieving the goal of sustainable operation (Manual on Material Flow Cost Accounting ISO14051, 2014).

Kokubu and Kitada (2015) investigated how the case companies used MFCA to introduce countermeasures to solve problems. The first case study indicated that Tanabe Seiyaku introduced MFCA as a trial in 2001. Although most Japanese companies still use Microsoft Excel for MFCA calculations, Tanabe Seiyaku has already combined the enterprise resource planning system for the entire company with MFCA (Kokubu, 2008). Through the integration of MFCA data, information on improvement activities is shared across the entire company. The second case is Canon Inc., one of the leading precise machine companies in Japan. The company introduced MFCA as a trial from 2001 to 2002, after which time Canon successively introduced MFCA at individual manufacturing plants on the basis of cooperation between the environmental department and plants. In a report on sustainable development (2005), Canon also mentioned that through its workplace-centered environmental assurance system using MFCA, they are committed to reducing waste and cost. The last case study focused on Sekisui Chemical Co. Ltd, which introduced MFCA in 2004 and indicated that through MFCA activities, the company continues to lower and stabilize costs, verifying that from a business perspective, using an innovative MFCA management system is conducive to the development of manufacturing industries.

Chompu-inwai et al. (2015) conducted a case study on a wood-product manufacturer in northern Thailand. An analysis of the company’s production process revealed that almost 70% of the raw wood materials used were waste in the form of chippings, sawdust, off-cuts, and defects. The objective of the study was to discuss whether the application of MFCA can reduce material consumption and waste to the maximum extent. The results of their experiment were subsequently integrated into practical applications, reducing the amount of wood material loss incurred during the cutting process from approximately 69% to 54%.[M W3]  The findings demonstrated that applying MFCA increased product quality and reduced the adverse environmental effects of the production process of the case-study company, thereby reducing cost and strengthening competitiveness.

In response to the rising cost of raw materials obtained from the natural environment, consistently high production costs, and the inevitable imposition of carbon or energy taxes by environmental agencies, the China Steel Corporation install  MFCA integrated with functions for managing resources, the environment, and waste. Findings obtained using this tool can serve as a basis for determining improvement options and provide clear insight into which types of cost and expenditure are the sources of maximum loss. Furthermore, these results reveal the flow and proportion of iron resources from main products, recycling and reuse within plants, and waste gases to facilitate the formulation of improvement options.

The aforementioned studies present cases of successful MFCA implementation. Thus, through the promotion of ISO 14051-based MFCA analysis, this study constructed a material flow model to determine the potential environmental and financial consequences of raw material and energy use and manufacturing processes. Other aims were to increase the transparency of material and energy use and manufacturing processes as well as highlight the loss of raw material input and waste-treatment cost that have always been neglected. Creating a management mechanism can effectively strengthen green designs and enhance the research and development of clean processes while also identifying methods for mitigating environmental effects and reducing costs, with the hope of achieving the optimal balance between the promotion of environmental conservation and economic growth.

In traditional cost accounting, the cost of finished goods increases with the number of steps in the production process, and this cost increases incrementally. The physical volume of materials do not necessary accumulate with the number of steps in a production process because they may be cut and lost during manufacturing, thereby generating enough waste whose quantities plus the final output equal the total physical volume of materials inputted.

In traditional cost accounting, the basic calculation model involves allocating manufacturing costs to various products. Therefore, the value of wastes generated during the production process in most companies is “zero.” Generally, little analysis is performed on waste, except for the amount of WTC recognized in environmental accounting.

The basic calculation model in MFCA involves using the same approach to allocate the manufacturing costs of finished goods (positive products) and waste (negative products) generated during the production process. Thus, the true value of generated waste (negative products) can be calculated, and a hotspot for improvement can be identified from this originally zero-value waste (negative products).

Methodology: is missing Please provide a methodology section. It seems you are employing a case study, but it is not presented until now: How, when and what data you have collected from this case?

Ans: We have added the Methodology section.

The results section is quite long in its length and must be better organized and focused because the flow of the discourse is difficult to follow.

Figure 1 Is not readable.

Ans: Thanks for reviewer’s suggestion. We have revised it as follows:

Linea 175: what do you mean with "strategically collaborate": Please reframe the sentence.

Ans: Thanks for reviewer’s suggestion. We have revised the sentence as follows:

Company T, established in 2001 with a capital of NT$15.3 billion, was a professional manufacturer and supplier of color filters and strategically collaborated with Company A in 2006.

185-190: reframe this sentence. It is not clear. Discussion: is missing Conclusions At the moment, conclusions are just a summary of the previous sections on results.

Ans: Thanks for reviewer’s suggestion. We have revised the sentence as follows:

To elucidate the environmental effects of its raw material and energy practices, Company T promoted the ISO14051-based MFCA in 2013, which could not only be used to comprehensively examine the transparency of the company regarding raw materials and energy use but also minimize its environmental effects while also reducing the costs incurred by losses. Ultimately, the company aimed to fulfill its social responsibilities in a low-interest-rate environment. Regarding research methods, we described the standard operating procedures for introducing MFCA analysis by using the “Introductory Manual for Industry Implementation of Material Flow Cost Accounting” from the Taiwan Ministry of Economic Affairs and the “Material Flow Cost Accounting Introduction.

Reviewer 3 Report

This paper focuses on Material Flow Cost Accounting in Waste Reduction. In general terms, the paper is well written, although it needs some clarifications, and the analysis is well developed. The main conclusion it reaches can be regarded as a basis for further research.

Comments:

1. The title of the paper seems to be too long. The introduction does not present the thesis / hypothesis of the article

2. There is a lack of in-depth meaning of corporate sustainability and its tools based on science literature,

The paper shows the imprecise use of the terms sustainability, CSR, environmental aspects, „social sustainability” (line XX-XX), although the definition of CSR and sustaianbility is given in the Introduction, it is not used along the paper.

I would suggest to consider to change the general context and the theoretical basis to corporate sustainability , and then sustainability accounting.

3. and there is also no convincing analysis of the importance and current use of Material Flow Cost Accounting:

- there is no comparison to other test results

- there is no reference to other cost accounting tools, what integration possibilities,

- there are no possibilities to integrate with other systems

4. In coclusions/suggestions there is no cost-benefit analysis of the proposal presented in the paper.

5. Figure 1 is not readable

6. It is not it is not obvious what it means „the regulatory aspects of corporate sustainable management”

Author Response

Dear reviewer, thanks for providing us these good suggestions. Followings are our replies.

This paper focuses on Material Flow Cost Accounting in Waste Reduction. In general terms, the paper is well written, although it needs some clarifications, and the analysis is well developed. The main conclusion it reaches can be regarded as a basis for further research.

1.  The title of the paper seems to be too long.

Ans: Thanks for reviewer’s suggestion. We have revised the title as “The Application of Material Flow Cost Accounting in Waste Reduction”

2.  The introduction does not present the thesis / hypothesis of the article

Ans: Thanks for reviewer’s suggestion. We have added the hypothesis in the Introduction.

The hypothesis of the research is whether the application of ISO 14051-based MFCA can assist case company to analyze the cost, which help manufacturing companies to evolve in a changing environment and economy.   

3. There is a lack of in-depth meaning of corporate sustainability and its tools based on science literature, The paper shows the imprecise use of the terms sustainability, CSR, environmental aspects, „social sustainability” (line XX-XX), although the definition of CSR and sustaianbility is given in the Introduction, it is not used along the paper. I would suggest to consider to change the general context and the theoretical basis to corporate sustainability, and then sustainability accounting.

Ans: Thanks for reviewer’s suggestion. We have deleted the CSR document in our paper because the other reviewer suggests that CSR is not useful for the aim of the paper and the contribution. Conversely, we focus more on material, environmental and performance accounting, that is much more interesting considering the results.

Environment protection is critical for sustainability. Continuous investments in energy consumption and natural resource consumption as well as manufacturing sectors and infrastructure have had seriously harmful impacts on the environment (Mahmood, Furqan & Bagais, 2019). Environmental accounting create accountability for business entities in terms of their efforts to protect the environment in their corporate decisions (Masud, Bae, & Kim, 2017). Environmental accounting constitutes a tool for applying the sustainable development concept and now commands acceptance as a means of ensuring the preservation of the environment Environmental accounting information includes financial, environmental performance, and policy aspects, which are scattered in many parts of annual reports and social responsibility reports. The quality of disclosures can be characterized from aspects of comprehensiveness, reliability, and compliance (Masud, Bae, & Kim, 2017). In decision making, an organization considers different pressures from internal and external parties and attempts to legitimize the impact of its activities on the environment in the eyes of society and various pressure groups. Environmental accounting plays an active role in preparing, presenting, and analyzing environmental information for interested parties holders, thus encouraging top management to improve environmental conditions (Masud, Bae, & Kim, 2017).

4. There is also no convincing analysis of the importance and current use of Material Flow Cost Accounting: - there is no comparison to other test results - there is no reference to other cost accounting tools, what integration possibilities, - there are no possibilities to integrate with other systems

Ans: Thanks for reviewer’s suggestion. We have added other test results to compare and discussed other cost accounting tools as follows:

Numerous studies worldwide have identified the advantages of MFCA. Based on a case study of a Sri Lankan crepe rubber factory, Dunuwila et al. (2018) adopted a cost-efficient and eco-friendly approach to improve the processing sector for natural rubber in the following three phases. 1. Quantifying factory resource use, economic loss, and greenhouse gas emissions through material flow analysis, MFCA, and life-cycle assessment [M W1] 2. developing proposals for viable improvement options; and 3. confirming the benefits of implementing the suggested improvement options. The results indicated that the proposed methods enabled 26% and 79% decreases in cost and global warming potential,[VS2]  respectively. Provided that the improvement options are viable, the results published by Dunuwila et al. (2018) can be used to ultimately increase profits for the rubber company, reduce the amount of toxic gas released into the atmosphere, reduce water consumption, and enhance the company image.

Kokubu and Kitada (2015) indicated that intensely competitive Japanese companies have identified further room for improvement by using MFCA because the loss concept in MFCA is different from the general concept of compliance in traditional business management. The value of MFCA-derived information is attributed to this concept of difference in loss.

Nitto is a leading Japanese diversified materials manufacturer that offers a wide variety of products, including tapes, vinyl, LCDs, insulation, and reverse osmosis membranes. With the support of the Japanese Ministry of Economy, Nitto was the first model company to use MFCA. Before introducing MFCA, Nitto has long been committed to enhancing the efficiency and mitigating the environmental effects of its operations. Nitto discovered that its environmental-accounting system was incomplete primarily because only environmental protection costs were considered, and a few costs associated with environmental effects were still hidden. The environmental protection costs incurred through conventional environmental accounting only represented 1.6% of total sales in 2005. However, other costs, such as energy-consumption costs related to environmental effects are usually regarded as production costs, which accounted for approximately 13.6% of total sales, substantially higher than that for environmental protection costs (1.6%). During the process of target-product selection, Nitto focused on a product that exhibited an upward market trend and had production lines that generated a considerable amount of material loss and consumed a substantial amount of energy during the manufacturing process. Through MFCA, Nitto could accurately assess its material losses and associated costs. This information became a driving force for Nitto to improve the material productivity of this production line. Hence, the implementation team established its improvement plans on the basis of a cost–benefit analysis using the physical and cost data under MFCA. The plans and countermeasures were subsequently planned and implemented. Nitto recognized MFCA as a method that was helpful for clarifying losses in terms of cost, enabled managers to set explicit improvement goals, and provided a means for identifying potential cost reductions and positive effects on the environment. Using a professional management method such as MFCA ensured increased competitiveness for achieving the goal of sustainable operation (Manual on Material Flow Cost Accounting ISO14051, 2014).

Kokubu and Kitada (2015) investigated how the case companies used MFCA to introduce countermeasures to solve problems. The first case study indicated that Tanabe Seiyaku introduced MFCA as a trial in 2001. Although most Japanese companies still use Microsoft Excel for MFCA calculations, Tanabe Seiyaku has already combined the enterprise resource planning system for the entire company with MFCA (Kokubu, 2008). Through the integration of MFCA data, information on improvement activities is shared across the entire company. The second case is Canon Inc., one of the leading precise machine companies in Japan. The company introduced MFCA as a trial from 2001 to 2002, after which time Canon successively introduced MFCA at individual manufacturing plants on the basis of cooperation between the environmental department and plants. In a report on sustainable development (2005), Canon also mentioned that through its workplace-centered environmental assurance system using MFCA, they are committed to reducing waste and cost. The last case study focused on Sekisui Chemical Co. Ltd, which introduced MFCA in 2004 and indicated that through MFCA activities, the company continues to lower and stabilize costs, verifying that from a business perspective, using an innovative MFCA management system is conducive to the development of manufacturing industries.

Chompu-inwai et al. (2015) conducted a case study on a wood-product manufacturer in northern Thailand. An analysis of the company’s production process revealed that almost 70% of the raw wood materials used were waste in the form of chippings, sawdust, off-cuts, and defects. The objective of the study was to discuss whether the application of MFCA can reduce material consumption and waste to the maximum extent. The results of their experiment were subsequently integrated into practical applications, reducing the amount of wood material loss incurred during the cutting process from approximately 69% to 54%.[M W3]  The findings demonstrated that applying MFCA increased product quality and reduced the adverse environmental effects of the production process of the case-study company, thereby reducing cost and strengthening competitiveness.

In response to the rising cost of raw materials obtained from the natural environment, consistently high production costs, and the inevitable imposition of carbon or energy taxes by environmental agencies, the China Steel Corporation install  MFCA integrated with functions for managing resources, the environment, and waste. Findings obtained using this tool can serve as a basis for determining improvement options and provide clear insight into which types of cost and expenditure are the sources of maximum loss. Furthermore, these results reveal the flow and proportion of iron resources from main products, recycling and reuse within plants, and waste gases to facilitate the formulation of improvement options.

One key difference is how you recognize the cost MFCA represents a different way of management accounting. In conventional cost accounting, the data is used to determine whether the incurred costs are recovered from sales. It does not require determining whether material is transformed into products, or disposed as waste. In conventional accounting, even if waste is recognized in terms of quantity, the costs to produce ‘material losses’ are included as part of the total output cost.

On the other hand, MFCA, as explained before, focuses on identifying and differentiating between the costs associated with ‘products’ and ‘material losses’. 

In this way ‘material loss’ is evaluated as an economic loss which encourages the management to search for ways to reduce material losses and improve business efficiency. The figure below shows the difference between MFCA and Conventional Accounting.

MFCA is the cost Accounting system can be installed in ERP system. Therefore, there is no system integration issue of MFCA system. 

4. In conclusions/suggestions there is no cost-benefit analysis of the proposal presented in the paper.

Ans: Thanks for reviewer’s suggestion. We have the cost-benefit analysis as follows.

The primary finding is that the film layer on recycled washed glass tends to be stripped during the production process, causing increased reprocessing costs and thus rendering the cost of renewable cleaning higher than that of reworking.

The case company utilizes MFCA to analyze the recycling process of film layer which can actually reflect the cost of the waste. The results can be applied to review the recycling process of the case company to effectively reduce the costs. The analysis can achieve the goal of reducing the costs and increasing the efficiency and set the model for promoting the sustainability.   

MFCA analysis tool is effectively optimizing the cost management system in reducing the cost especially when the limited resources are input by the business. Thus, the implementation of MFCA analysis tool receives more benefit than cost spending.

5. Figure 1 is not readable

Ans: Thanks for reviewer’s suggestion. We have revised Figure 1 as follows:

 6. It is not it is not obvious what it means „the regulatory aspects of corporate sustainable management

Ans: Thanks for reviewer’s suggestions. We have deleted this sentence.

Round  2

Reviewer 1 Report

Sustainability

Second review report of

Manuscript nº 423048: “The Application of Material Flow Cost Accounting in Waste Reduction”

 The authors did not respond appropriately to my review comments. In my previous suggestions I said that “the paper is mainly unclear in terms of objectives. It seems a descriptive study and not an analytical case study. Case studies usually respond to ‘How’ and ‘Why’ questions. The research method is also not well explained.”

 The authors introduced a research method section as a response to my comments. But this section by itself does not respond to my concerns: it does not help the reader to understand the paper. The research question is not presented. What is the gap in the literature that this paper is intending to fill? At the moment, it seems the paper is just “an example to analyze the implementation of ISO 14051-based MFCA by the case company and investigate the results” (p.8). Further, the authors state that they “also have conducted an in-depth interview with the production manager, accounting manager and other internal personnel to further understand the flow of the company's existing product line and cost structure” (p. 9), but the interviewees were not presented, and readers will be unable to understand how the interviews contributed to the paper. The interviews are not mentioned in the results section.

 Additionally, the authors did not prepare this section carefully. They repeat this paragraph: “Although the case-study method provides a comprehensive understanding of a situation and causal relationships, it is still overly subjective and lacks objectivity. Thus, in our case analysis, we first described the implementation manuals adopted in Taiwan and and then analyzed the case company by following the implementation methods specified by the manual to circumvent the effects of subjectivity on this research” (both on page 9).

 The tables are now introduced to the reader. However, there is no theory. There is merely a technical practical example of the implementation of ISO 14051-based MFCA. Thus, we do not have a research problem but a practical problem.

 My concern about contribution remains. The authors argue that “The study promotes the application of MFCA to SMEs in Malaysia because the case company sets an example to other SMEs that MFCA can be embarked on with relative ease, and indicates perhaps an inexpensive cost of implementation” (p. 23). A case study in Taiwan contributes to SMEs in Malaysia? Or the cited reference [6] already did the contribution the authors want to do? I wonder whether the ISO 14051-based MFCA does not provide guidelines for implementation? We should also assume that if it is being advised to companies by the ISO, it is because it is easy and feasible.

 As I mentioned before,I cannot consider this paper to be a research paper. It is a technical case study. Perhaps an important research questions that the authors could develop is: Why did the case company successfully implement the MFCA? And how was it implemented? I understand that this is the first implementation in Taiwan and it would be interesting to understand the reasons for early and voluntary adoption. Case studies use “How” and Why” research questions and aim to extend theory. I cannot see this in this paper.

Author Response

Response to Reviewer 1 Comments

Point 1: The authors did not respond appropriately to my review comments. In my previous suggestions I said that “the paper is mainly unclear in terms of objectives. It seems a descriptive study and not an analytical case study. Case studies usually respond to ‘How’ and ‘Why’ questions. The research method is also not well explained.

Response 1: Thanks for reviewer’s suggestion. We have revised the part as follows:

      Taiwan has serious environment pollution problem such as air pollution caused by the power plants and pollution of water sources, which has a significant impact on the society, people and enterprises. Even though many SMEs have the environmental awareness, they do not have the knowledge required and/or the skills needed to apply the appropriate environmental tools  [29,31].One study indicated that the eco-efficient optimization of material flow cost management, which aims to reduce costs while simultaneously decreasing environmental influence, has become an explicit objective of practical and scientific activities and efforts. Although discussions have been frequent, little has actually been done to realize these goals [11]. Even in Germany and Japan, numerous companies remain ill-informed regarding the MFCA process and few have fully incorporated this practice into their operations [35]. However, Schaltegger et al. [30] used tools for sustainability management to examine large German companies and reported that more well-known environmental accounting tools are more likely to be implemented. Therefore, communication channels and publicity regarding the release of the ISO 14051: Material flow cost-accounting standard are crucial [35]. 

       To standardize MFCA practices, a working group of the ISO Technical Committee ISO/ TC 207, Environmental Management, developed ISO 14051, which complements the ISO 14000 family of environmental management system standards, including life cycle assessment (ISO 14040, ISO 14044) and environmental performance evaluation (ISO 14031). The standard was published in the second half of 2011. Material flow cost accounting (MFCA) based on ISO 14051 is an environmental-management accounting (EMA) tool providing the analysis of environmental and economic balance [3]. ISO 14051 provides a general framework for material flow cost accounting. The ultimate goal of MFCA is to mitigate environmental problems and simultaneously improve economic performance [4]. The concept of MFCA has been successfully applied in numerous industrial practices to improve waste-reduction decisions such as brewing companies in South Africa [3], small textile factories in Thailand [5], pharmaceutical companies in Japan [4], and small and medium-sized enterprises in Malaysia [6]. These cases indicate that MFCA increases environmental sustainability and improves the overall economic performance of a company.

       The purpose of this study is to use case companies to prove whether the introduction of MFCA management tools can be successfully promoted only by referring to ISO14501 manual. In addition, by using the financial and cost data provided by the case company, this study explores whether MFCA analysis tool is a set of stability tools, and whether it is possible to produce different results because of the different managers of operation analysis tools.The case company is the first company in Taiwan fully implementing MFCA. The reason why the case company takes the lead to implement MFCA is to reduce the waste and pollution, which can assist the case company to reach the sustainability. Other company in Taiwan only implement the MFCA in one factory or one product line which is difficult to know real effects of the implementation. Thus, to understand the implementation process and steps of the case company can provide the suggestions to other companies.

Point 2: The authors introduced a research method section as a response to my comments. But this section by itself does not respond to my concerns: it does not help the reader to understand the paper. The research question is not presented. What is the gap in the literature that this paper is intending to fill? At the moment, it seems the paper is just “an example to analyze the implementation of ISO 14051-based MFCA by the case company and investigate the results” (p.8). Further, the authors state that they “also have conducted an in-depth interview with the production manager, accounting manager and other internal personnel to further understand the flow of the company's existing product line and cost structure” (p. 9), but the interviewees were not presented, and readers will be unable to understand how the interviews contributed to the paper. The interviews are not mentioned in the results section.

Response 2: We have revised the part in interviews as follow:

      According to Clause 6.2, 6.3, ISO 14051:2011, successful projects usually start with the supports of the company’s management. MFCA is not an exception. Through engagement of the company management in the MFCA implementation process, the required expertise can be determined and the correct flow of information between all areas involved can be facilitated. Therefore, we also have in-depth interview with production manager, accounting manager and other internal personnel to further understand the flow of the company's existing product line and cost structure. We interview five employees including one production manager, one account manager, three internal staffs to increase the integrity of data collection through discussion and observation of visits.

Response 2:Thanks for reviewer’s suggestion. We have revised the gap in the literature. We have added the 28th reference as follow:

28.        Gosman, M. L., and Kohlbeck, M. J. Effects of the existence and identity of major customers on supplier profitability: Is Wal-Mart different? J. Manage. Account.Rres.  2009, 21, 179-201.

29.        Daian, G.; Ozarska, B., Wood waste management practices and strategies to increase sustainability standards in the Australian wooden furniture manufacturing sector. J .Clean. Prod.  2009, 17 (17), 1594-1602.

30.        Schaltegger, S.; Windolph, S. E.; Herzig, C. From knowledge to application: Dissemination of sustainability management tools in large german companies. Centre for Sustainability Management:Leuphana University.Lueneburg, 2011.

31.        Van Hoof, B.; Lyon, T. P., Cleaner production in small firms taking part in Mexico's Sustainable Supplier Program. J .Clean. Prod. 2013, 41, 270-282.

32.        Yin, R. K. Case Study Research: Design and Methods, 2nd ed; Thousand Oaks,CA: Sage Publications. 1994.

33.        Banker, R. D., Potter, G., and Srinivasan, D. An empirical investigation of an incentive plan that includes nonfinancial performance measures. Account. Rev. 2000, 75, 65-92.

34.        Strobel, M., Redmann, C. Flow cost accounting, an accounting approach based on the actual flows of materials. In: Bennett, M., Bouma, J.J., Walters, T. (Eds.); Environmental Management Accounting: Informational and Institutional Developments; Kluwer: Dordrecht, Netherlands, 2002; pp. 67-82.

35.        Christ, K. L.; Burritt, R. L., ISO 14051: A new era for MFCA implementation and research.  Revista de Contabilidad - Spanish Accounting Review. 2016, 19, 1-9.

Point 3: Additionally, the authors did not prepare this section carefully. They repeat this paragraph: “Although the case-study method provides a comprehensive understanding of a situation and causal relationships, it is still overly subjective and lacks objectivity. Thus, in our case analysis, we first described the implementation manuals adopted in Taiwan and then analyzed the case company by following the implementation methods specified by the manual to circumvent the effects of subjectivity on this research” (both on page 9). 

Response 3: Thanks for reviewer’s suggestion. We have revised the part such as point 1.

      Taiwan has serious environment pollution problem such as air pollution caused by the power plants and pollution of water sources, which has a significant impact on the society, people and enterprises. Even though many SMEs have the environmental awareness, they do not have the knowledge required and/or the skills needed to apply the appropriate environmental tools [29,31].One study indicated that the eco-efficient optimization of material flow cost management, which aims to reduce costs while simultaneously decreasing environmental influence, has become an explicit objective of practical and scientific activities and efforts. Although discussions have been frequent, little has actually been done to realize these goals [11]. Even in Germany and Japan, numerous companies remain ill-informed regarding the MFCA process and few have fully incorporated this practice into their operations [35]. However, Schaltegger et al. [30] used tools for sustainability management to examine large German companies and reported that more well-known environmental accounting tools are more likely to be implemented. Therefore, communication channels and publicity regarding the release of the ISO 14051: Material flow cost-accounting standard are crucial [35].

      To standardize MFCA practices, a working group of the ISO Technical Committee ISO/ TC 207, Environmental Management, developed ISO 14051, which complements the ISO 14000 family of environmental management system standards, including life cycle assessment (ISO 14040, ISO 14044) and environmental performance evaluation (ISO 14031). The standard was published in the second half of 2011. Material flow cost accounting (MFCA) based on ISO 14051 is an environmental-management accounting (EMA) tool providing the analysis of environmental and economic balance [3]. ISO 14051 provides a general framework for material flow cost accounting. The ultimate goal of MFCA is to mitigate environmental problems and simultaneously improve economic performance [4]. The concept of MFCA has been successfully applied in numerous industrial practices to improve waste-reduction decisions such as brewing companies in South Africa [3], small textile factories in Thailand [5], pharmaceutical companies in Japan [4], and small and medium-sized enterprises in Malaysia [6]. These cases indicate that MFCA increases environmental sustainability and improves the overall economic performance of a company.

      The purpose of this study is to use case companies to prove whether the introduction of MFCA management tools can be successfully promoted only by referring to ISO14501 manual. In addition, by using the financial and cost data provided by the case company, this study explores whether MFCA analysis tool is a set of stability tools, and whether it is possible to produce different results because of the different managers of operation analysis tools.

The case company is the first company in Taiwan fully implementing MFCA. The reason why the case company takes the lead to implement MFCA is to reduce the waste and pollution, which can assist the case company to reach the sustainability. Other company in Taiwan only implement the MFCA in one factory or one product line which is difficult to know real effects of the implementation. Thus, to understand the implementation process and steps of the case company can provide the suggestions to other companies.

Point 4: The tables are now introduced to the reader. However, there is no theory. There is merely a technical practical example of the implementation of ISO 14051-based MFCA. Thus, we do not have a research problem but a practical problem.

Response 4: Thanks for reviewer’s suggestion. We have revised the section as follows:

3. Research Method

This research adopted the case-study method using a flat-panel parts supplier as an example to analyze the implementation of ISO 14051-based MFCA by the case company and investigate the results. This method is most commonly used in exploratory research for which the relevant qualitative research techniques are convenient [17]. Case study method can disclosure the problem and then systematically collect and analyze the problem in a scientific way to find the solution [32]. The importance of this case study is the process of applying MFCA can provide the suggestions for future installation of MFCA. For example, the setting of the quantity center should be the same as the existing data record and periodic review framework to avoid the ineffective use. Second, the historical data should be saved in the operating system, and the material flow currency should be quantitatively linked to the accounting, and the existing management system of each department should be comprehensively integrated to facilitate the rapid introduction of product. The analysis of material flow and energy use assists company understand the relationship between material usage and cost, identify the hot spots of material and cost loss, improve the efficiency of data analysis and systematically review the process to promote and strengthen the environmental cost management.

Point 5: My concern about contribution remains. The authors argue that “The study promotes the application of MFCA to SMEs in Malaysia because the case company sets an example to other SMEs that MFCA can be embarked on with relative ease, and indicates perhaps an inexpensive cost of implementation” (p. 23). A case study in Taiwan contributes to SMEs in Malaysia? Or the cited reference [6] already did the contribution the authors want to do? I wonder whether the ISO 14051-based MFCA does not provide guidelines for implementation? We should also assume that if it is being advised to companies by the ISO, it is because it is easy and feasible.

Response 5: Thanks for reviewer’s suggestions.  Due to the typo, we have revised the paragraph as follows: 

      By observing the case company's promotion of the ISO14051 MFCA tool, we can confirm that the inter-departmental flow of information is more accurate according to the steps in Chapter 2.3.1, which helps organizations better understand the application of materials and energy. Because the scope of sharing was expanded, it helped to improve the company to reduce the implementation of operating procedures. Through this study, we also found that managers could understand the MFCA tool more easily, and also helped the organization to better understand the application of materials and energy. That is to say, even if different managers operate MFCA tools, the results of this study are not very different from those calculated by the case company. Therefore, it is an analytical management tool that can be popularized.

      MFCA analysis tool is effectively optimizing the cost management system in reducing the cost especially when the limited resources are input by the business. Thus, the implementation of MFCA analysis tool receives more benefit than cost spending. The academic contributions of the study are as follows. The study promotes the application of MFCA to SMEs because the case company sets an example to other SMEs that MFCA can be embarked on with relative ease, and indicates perhaps an inexpensive cost of implementation. The research fills the research gap by providing the steps and suggestions of the successful application of ISO 14051 based MFCA case in Taiwan. The case company utilizes MFCA to analyze the recycling process of film layer which can actually reflect the cost of the waste. The results can be applied to review the recycling process of the case company to effectively reduce the costs. The analysis can achieve the goal of reducing the costs and increasing the efficiency and set the model for promoting the sustainability.

Point 6: As I mentioned before,I cannot consider this paper to be a research paper. It is a technical case study. Perhaps an important research questions that the authors could develop is: Why did the case company successfully implement the MFCA? And how was it implemented? I understand that this is the first implementation in Taiwan and it would be interesting to understand the reasons for early and voluntary adoption. Case studies use “How” and Why” research questions and aim to extend theory. I cannot see this in this paper.

Response 6: Thanks for reviewer’s suggestion. We have revised the part such as point 1, point 2.

      Taiwan has serious environment pollution problem such as air pollution caused by the power plants and pollution of water sources, which has a significant impact on the society, people and enterprises. Even though many SMEs have the environmental awareness, they do not have the knowledge required and/or the skills needed to apply the appropriate environmental tools [29,31].One study indicated that the eco-efficient optimization of material flow cost management, which aims to reduce costs while simultaneously decreasing environmental influence, has become an explicit objective of practical and scientific activities and efforts. Although discussions have been frequent, little has actually been done to realize these goals [11]. Even in Germany and Japan, numerous companies remain ill-informed regarding the MFCA process and few have fully incorporated this practice into their operations [35]. However, Schaltegger et al. [30] used tools for sustainability management to examine large German companies and reported that more well-known environmental accounting tools are more likely to be implemented. Therefore, communication channels and publicity regarding the release of the ISO 14051: Material flow cost-accounting standard are crucial [35].

      To standardize MFCA practices, a working group of the ISO Technical Committee ISO/ TC 207, Environmental Management, developed ISO 14051, which complements the ISO 14000 family of environmental management system standards, including life cycle assessment (ISO 14040, ISO 14044) and environmental performance evaluation (ISO 14031). The standard was published in the second half of 2011. Material flow cost accounting (MFCA) based on ISO 14051 is an environmental-management accounting (EMA) tool providing the analysis of environmental and economic balance [3]. ISO 14051 provides a general framework for material flow cost accounting. The ultimate goal of MFCA is to mitigate environmental problems and simultaneously improve economic performance [4]. The concept of MFCA has been successfully applied in numerous industrial practices to improve waste-reduction decisions such as brewing companies in South Africa [3], small textile factories in Thailand [5], pharmaceutical companies in Japan [4], and small and medium-sized enterprises in Malaysia [6]. These cases indicate that MFCA increases environmental sustainability and improves the overall economic performance of a company.

      The purpose of this study is to use case companies to prove whether the introduction of MFCA management tools can be successfully promoted only by referring to ISO14501 manual. In addition, by using the financial and cost data provided by the case company, this study explores whether MFCA analysis tool is a set of stability tools, and whether it is possible to produce different results because of the different managers of operation analysis tools. The case company is the first company in Taiwan fully implementing MFCA. The reason why the case company takes the lead to implement MFCA is to reduce the waste and pollution, which can assist the case company to reach the sustainability. Other company in Taiwan only implement the MFCA in one factory or one product line which is difficult to know real effects of the implementation. Thus, to understand the implementation process and steps of the case company can provide the suggestions to other companies.

      According to Clause 6.2, 6.3, ISO 14051:2011, successful projects usually start with the supports of the company’s management. MFCA is not an exception. Through engagement of the company management in the MFCA implementation process, the required expertise can be determined and the correct flow of information between all areas involved can be facilitated. Therefore, we also have in-depth interview with production manager, accounting manager and other internal personnel to further understand the flow of the company's existing product line and cost structure. We interview five employees including one production manager, one account manager, three internal staffs to increase the integrity of data collection through discussion and observation of visits.

Reviewer 2 Report

Dear Authors,

Thank you for giving me the possibility to review your revised paper that is much improved from the first round.

Some additional review is needed in order to render this paper publishable.

Please see my further comments below.

Line 287 Please start a new section here on the ISO and use the parts in block letters related on the 5 steps. These could be subsections.

Research method sounds a bit done as homework. It is not necessary you spend from line 378 to 399 for justifying the use of the case study methodology in managerial studies. This part is redundant: Try to be more effective here. For example, move the last paragraph before line 400.

Line 415: how many interviews did you do?

Lines 427-439: do not repeat the list. Just state that you are presenting the data according to the steps presented in section XX.

Page 15 line 582: you refer to “two types of od work orders”, but the following list is having three points.

Page 18: Insert the new part in a table or in the appendix

I am bringing back my previous comment: “Contribution seems clear but is not supported by a literature gap and a research aim. A research question will help you to create your research framework”

The discussion is still missing. Section 5 is still not connected with the previous literature. Please, pick your literature review sections and try to couple your results, summarized in the discussion, with the previous studies, underlying what you are stating of similar or different.

The paper must be polished by a scientific editing service. It is plenty of typing mistake, isolate sentences and other formal mistake not in compliance with the journal format.

Reference from the 28th  are not available!!!

The paper is much improved, but some further steps are needed. I hope my comments will help you.

Author Response

Response to Reviewer 2 Comments

Point 1: Line 287 Please start a new section here on the ISO and use the parts in block letters related on the 5 steps. These could be subsections.

Response 1: Thanks for reviewer’s suggestion. We have revised the part as follows:

2.3. ISO 14051 based MFCA

       ISO 14051 defines MFCA as a “tool for quantifying the flows and stocks of materials in processes or production lines in both physical and monetary units” where ‘materials’ include energy and water [9]. These flows and stocks of materials are important because they pervade business practice [19]. The aim of MFCA is to provide information to management about opportunities for reducing materials use and improving monetary performance of businesses at the same time, an irresistible opportunity.

        To facilitate implementation, ISO 14051 proposes several MFCA implementation steps as delineated below. The level of detail and complexity of the analysis will depend on the size of the organization, nature of the organization’s activities and products, number of processes, and quantity centers chosen for analysis, among other factors. These conditions make MFCA a flexible tool that can be applied in a wide range of organizations.

2.3.1 Implementation step 1: Engaging management and determining roles and responsibilities (Clause 6.2, 6.3, ISO 14051:2011)

       Successful projects usually start with support from the company’s management; MFCA is not an exception. If the company management understands the benefits of MFCA and its usefulness in achieving an organization’s environmental and financial targets, it is easier to gain commitment from the whole organization. In order to be effectively implemented, it is highly recommended that top management take the lead in MFCA implementation by assigning roles and responsibilities, including setting up an MFCA project implementation team, providing resources, monitoring progress, reviewing results, and deciding on improvement measures based on the MFCA results.

       In general terms, management should be engaged in all phases of MFCA implementation and it is recommended that MFCA projects start with aggressive support from management, followed by a bottom-up approach on-site. In addition, successful implementation of MFCA requires collaboration between different departments within the organization. The reason why collaboration is needed is because different sources of information are required to complete MFCA analysis. Through engagement of the company management in the MFCA implementation process, the required expertise can be determined and the correct flow of information between all areas involved can be facilitated.

2.3.2 Implementation step 2: Scope and boundary of the process and establishing a material flow model (Clauses 6.4, 6.5, ISO 14051:2011)

       Based on collected material flow data, the MFCA boundary needs to be specified to understand clearly the scale of MFCA activity. During implementation, it is usually recommended to focus on specific products or processes at the beginning and then expanding implementation to other products. By implementing MFCA in steps, the analysis is simplified and better results can be achieved.

       The boundary can be limited to a single process, multiple processes, an entire facility, or a supply chain. It is recommended that the process or processes that are chosen for initial implementation be the ones with potentially significant environmental and economic impacts. After specifying the boundary, the process should be classified in quantity centers using process information and procurement records. In MFCA, the quantity center is the part of the process in which inputs and outputs are quantified. In most cases, quantity centers represent parts of the process in which materials are transformed. If the material flow between two processes is the source of significant material loss, the flow can be classified as a separate material flow.

        After determining the boundary and quantity centers, a time period for MFCA data collection needs to be specified. While MFCA does not indicate the period during which data must be collected for analysis, it should be sufficiently long to allow meaningful data to be collected and to minimize the impact of any significant process variation that can affect the reliability and usability of the data, such as seasonal fluctuations. Several historical MFCA projects indicate that the appropriate data collection period can be as short as a month, with a half-year or a year of data collection being the most common.

2.3.3 Implementation step 3: Cost allocation (Clauses 5.3, 6.8, ISO 14051:2011)

MFCA divides costs into the following categories:

          Material cost: cost for a substance that enters and/or leaves a quantity center

          Energy cost: cost for electricity, fuel, steam, heat, and compressed air

          System cost: Cost of labor, cost of depreciation and maintenance, and cost of transport

          Waste management cost: cost of handling waste generated in a quantity center

       Material costs, energy costs, and system costs are assigned or allocated to either products or material losses at each quantity center based on the proportion of the material input that flows into product and material loss. The material costs for each input and output flow are quantified by multiplying the physical amount of the material flow by the unit cost of the material over the time period chosen for the analysis. When quantifying the material costs for products and material losses, the material costs associated with any changes in material inventory within the quantity center should also be quantified. In contrast to material, energy, and system costs that are assigned to products and material losses proportionally, 100% of the waste management costs are attributed to material loss, since the costs represent the costs of managing this material loss.

2.3.4 Implementation step 4: Interpreting and communication MFCA results (Clauses 6.9, 6.10, ISO 14051:2011)

       MFCA implementation provides information such as material loss throughout the process, the use of materials that do not become products, overall costs, and energy and system costs associated with the material loss. This information brings about multiple impacts by increasing the awareness of the company’s operations. Managers who are aware of the costs associated with material losses can then identify opportunities to increase efficiency in material use and improve business performance.

2.3.5 Implementation step 5: Improving production practices and reducing material loss through MFCA results (Clause 6.11, ISO 14051:2011)

       Once MFCA analysis has assisted an organization to understand the magnitude, consequences, and drivers of material use and loss, the organization may review the MFCA data and seek opportunities to improve environmental and financial performance. The measures taken to achieve these improvements can include substitution of materials; modification of processes, production lines, or products; and intensified research and development activities related to material and energy efficiency.

Point 2: Research method sounds a bit done as homework. It is not necessary you spend from line 378 to 399 for justifying the use of the case study methodology in managerial studies. This part is redundant: Try to be more effective here. For example, move the last paragraph before line 400.

Response 2: Thanks for reviewer’s suggestion. We have revised the section as follows: 

3. Research Method

       This research adopted the case-study method using a flat-panel parts supplier as an example to analyze the implementation of ISO 14051-based MFCA by the case company and investigate the results. This method is most commonly used in exploratory research for which the relevant qualitative research techniques are convenient [17]. Case study method can disclosure the problem and then systematically collect and analyze the problem in a scientific way to find the solution [32]. The importance of this case study is the process of applying MFCA can provide the suggestions for future installation of MFCA. For example, the setting of the quantity center should be the same as the existing data record and periodic review framework to avoid the ineffective use. Second, the historical data should be saved in the operating system, and the material flow currency should be quantitatively linked to the accounting, and the existing management system of each department should be comprehensively integrated to facilitate the rapid introduction of product. The analysis of material flow and energy use assists company understand the relationship between material usage and cost, identify the hot spots of material and cost loss, improve the efficiency of data analysis and systematically review the process to promote and strengthen the environmental cost management.

Point 3: Line 415: how many interviews did you do?

Response 3: We interview five employees including one production manager, one account manager, three internal staffs.

Point 4: Lines 427-439: do not repeat the list. Just state that you are presenting the data according to the steps presented in section XX. 

Response 4: Thanks for reviewer’s suggestion. We have revised the paper as follows: 

      To facilitate implementation, ISO 14051 proposes five MFCA implementation steps as delineated in the literature review section, which are listed in section 2.3.

Point 5: Page 15 line 582: you refer to “two types of od work orders”, but the following list is having three points.

Response 5: Thanks for reviewer’s suggestion. The correction description should only have two type. We have move the third point to the Appendix.

       The two types of work orders are described as follows:

(1)        Standard-order bill of material: A standard-order bill of material is the and also consists of the amount of materials used in each quantity center, hours of labor, and other manufacture quantity of input materials required to produce a piece of color filter without any substantial loss ring costs.

(2)        Production-order bill of material: It is almost impossible that no loss incurred during the production process. Therefore, production-management staff adopt a standard-order bill of material as the basis for adjusting input quantity according to the status of a production line. Adjustments are most commonly made to yields. On a production-order bill of material, the quantity of usage is also the basis for calculating material stock and the quantity of materials to be purchased. Cost accounting is based on the data provided on a production-order bill of material, which documents production start time, actual usage in each quantity center, and the amount of loss. In the case of actual usage, cost is calculated by multiplying bill of material-based input quantities in each quantity center by the standard unit price. Standard unit price is the unit price of a most recently approved order. The actual cost of each quantity center is summarized in Table 2. In the following section, adjustments to figures in the table are simulated. After a work order has been completed and inventoried, a work order record sheet is completed, as displayed in Table 3.

Point 6: Page 18: Insert the new part in a table or in the appendix

Response 6: Thanks for reviewer’s suggestion. We have move the part to Appendix A.

Point 7: I am bringing back my previous comment: “Contribution seems clear but is not supported by a literature gap and a research aim. A research question will help you to create your research framework”

Response 7: Thanks for reviewer’s suggestion. We have added a paragraph to strengthen the contribution. 

      The academic contributions of the study are as follows. The study promotes the application of MFCA to SMEs because the case company sets an example to other SMEs that MFCA can be embarked on with relative ease, and indicates perhaps an inexpensive cost of implementation. The research fills the research gap by providing the steps and suggestions of the successful application of ISO 14051 based MFCA case in Taiwan. The case company utilizes MFCA to analyze the recycling process of film layer which can actually reflect the cost of the waste. The results can be applied to review the recycling process of the case company to effectively reduce the costs. The analysis can achieve the goal of reducing the costs and increasing the efficiency and set the model for promoting the sustainability.

Point 8: The discussion is still missing. Section 5 is still not connected with the previous literature. Please, pick your literature review sections and try to couple your results, summarized in the discussion, with the previous studies, underlying what you are stating of similar or different.

Response 8: Thanks for reviewer’s suggestion. We have revised the paper to strengthen the conclusion.  

      Companies have begun to increasingly focus on how they can align requirements for social sustainability and competitiveness [20]. When companies cease to pursue only individual sustainable operations, they inevitably regard the environment as a component of corporate development [21]. MFCA has expanded the scope of conventional accounting to include the environment and the economic effects of corporate sustainability and eco-efficiency concepts [22,15], thereby rendering goal management increasingly more scientific and systematic. This study assists companies to examine the process of recycled glass production by using ISO 14051-based MFCA and investigate whether company can reduce costs by using recycled glass and mitigate the pollution and improve the companies’ environmental performances. 

      There is no previous paper discussing the MFCA implementation steps and providing the suggestions in reducing the cost in Taiwan. One of the reason is that the number of certified companies in Taiwan remains low, and whether these companies continue to conduct analyses after receiving certification is unknown. The reason that case company successfully implement MFCA is the attitude of its chief executive officer. In the second year of the implementation, the case company performed an analysis of the entire plant and discovered that its wastewater-treatment expenses were positively correlated with resist loss, chemical usage, and water consumption. In one month that did not exhibit a positive correlation, the company discovered that the sluice gate controlling wastewater discharge was damaged, which was causing a considerable increase in wastewater-treatment expenses because of the direct discharge of untreated wastewater. Although MFCA does not yield immediate effects regarding increase in company profits, this practice demonstrates another type of long-term corporate value that contributes to effective cost reduction in low-profit environments. Comparing with the previous research, this paper not only introduce the advantages and effects of implementing the MFCA bust also list the key action plan when applying the MFCA. This assists future company to have a complete idea in assessing whether to implement the MFCA. 

      To bolster the competitiveness, companies have continually invested monetary resources in the adoption of environmentally friendly practices. However, questions are often raised during this process regarding whether such an investment is truly beneficial for the environment or corporate development, and some companies remain focused on acquiring certifications to boost sales performance. Currently, introducing MFCA helps companies to obtain not only certificates that contribute to sales but also rewards in the form of cost reduction due to environmental initiatives.

     Team composition, interpersonal communications, and the efforts of revolutionists are factors that contribute to the success of the MFCA implementation. However, a potential barrier for improving MFCA is the inability of senior managers to solve performance-management problems [6]. Government authorities in Taiwan have begun to follow the steps implemented in Japan in terms of encouraging companies to introduce ISO14051-based MFCA analysis. Personnel in contact with companies primarily include those from environmental-safety, quality-control, or plant affairs departments. However, the basic information required for the analysis is generally managed by the finance and accounting departments. Therefore, the implementation of MFCA should collect, analyze, compile, and describe data from the perspectives of financial and accounting specialists to alert these specialists to the difference between conventional cost accounting and MFCA. After a balance sheet for basic analysis has been obtained, it should then be administered to plant personnel for hotspot improvements. 

      We determined that the original rate of product yield for the case company was 99.1%, indicating that the costs of negative products were low. Of all costs, the cost of glass materials accounted for 51% of material cost (67.1%). During analysis, we discovered that reducing the glass-stripping rate resulted in a reduction of other material, energy, and system costs required for reworking. However, recycled glass is associated with a high stripping rate. Whether the sum of this cost and the cost of renewable cleaning is lower than the cost of reworking should be re-evaluated to determine the benefits of outsourcing rework. The analysis also indicated a loss rate of 3.09% in energy cost. In response, the company can improve its power supply by installing smart electricity meters and developing energy-resource management systems to monitor energy usage. Therefore, this study verified that MFCA analysis both provides insights regarding areas that require improvement in the internal process management of the company and helps the company to achieve sustainable development with the goal of promoting environmental sustainability and reducing emissions.

Point 9: The paper must be polished by a scientific editing service. It is plenty of typing mistake, isolate sentences and other formal mistake not in compliance with the journal format.

Response 9: Thanks for reviewer’s suggestions. We already have the professional editing service to polish the paper. 

Point 10: Reference from the 28th  are not available!!!

Response 10:  Thanks for reviewer’s suggestion. We have add the 28th reference as follow

28.        Gosman, M. L., and Kohlbeck, M. J. Effects of the existence and identity of major customers on supplier profitability: Is Wal-Mart different?. J. Manage. Account.Rres.  2009, 21, 179-201.

29.        Daian, G.; Ozarska, B., Wood waste management practices and strategies to increase sustainability standards in the Australian wooden furniture manufacturing sector. J .Clean. Prod.  2009, 17 (17), 1594-1602.

30.        Schaltegger, S.; Windolph, S. E.; Herzig, C. From knowledge to application: Dissemination of sustainability management tools in large german companies. Centre for Sustainability Management:Leuphana University.Lueneburg, 2011.

31.        Van Hoof, B.; Lyon, T. P., Cleaner production in small firms taking part in Mexico's Sustainable Supplier Program. J .Clean. Prod. 2013, 41, 270-282.

32.        Yin, R. K. Case Study Research: Design and Methods, 2nd ed; Thousand Oaks,CA: Sage Publications. 1994.

33.        Banker, R. D., Potter, G., and Srinivasan, D. An empirical investigation of an incentive plan that includes nonfinancial performance measures. Account. Rev. 2000, 75, 65-92.

34.        Strobel, M., Redmann, C. Flow cost accounting, an accounting approach based on the actual flows of materials. In: Bennett, M., Bouma, J.J., Walters, T. (Eds.); Environmental Management Accounting: Informational and Institutional Developments; Kluwer:Dordrecht, Netherlands, 2002; pp. 67-82.

35.        Christ, K. L.; Burritt, R. L., ISO 14051: A new era for MFCA implementation and research.  Revista de Contabilidad - Spanish Accounting Review. 2016, 19, 1-9.

Round  3

Reviewer 1 Report

The paper eliminated some mistakes but in my view it is too descriptive.

Reviewer 2 Report

Dear authours,

I have really appreciated your great efforts for improving the quality of your paper that, in my opinion, is ready to be pubblishable on Sustainability!

Good luck with your future research

Best